# Complex marine ecological response during the Eocene-Oligocene revealed by global foraminiferal record

Zhengbo Lu [1,2,11], Ke Xue [3,4,11], Yiying Deng [5,11], Junxuan Fan [1,2] ✉, Peiyue Fang [6], Bridget S. Wade [7], Laia Alegret[8], Michael J. Benton [9], Yuchang Wu [3,4], Chao Qian [3,4] ✉, Xudong Hou [1,2], Yukun Shi [1,2] ✉, Peter M. Sadler [10], Huiqing Xu [1,2], Zhi-Hua Zhou [3,4] & Shuzhong Shen [1,2]

The Eocene–Oligocene transition was the crucial turning point when Earth's climate shifted to its current icehouse state. Understanding how the marine biosphere responded during this transition is not well-constrained, appearing as a simple extinction pulse in low temporal resolution global compendia. Here we design an artificial-intelligence-inspired metaheuristics algorithm to construct a high-resolution global species richness history across the Eocene–Oligocene transition for the rich foraminifera fossil record with an imputed ~29,000-year resolution. The revealed diversity dynamics are complex and differ for each foraminiferal group with distinct ecology. Planktonic and shallow-water larger benthic foraminifera show steady diversity levels in the early phases of the transition in the latest Eocene after a long-term reduction, while the deeper-water small benthic foraminifera radiate notably and then decline over the same interval. In the earliest Oligocene, the planktonic and larger foraminifera suffer major species losses coincident with the first continental-scale ice sheet formed on Antarctica, while small benthic foraminifera diversity holds steady, followed by an accelerating lowering as the early Oligocene proceeds. These findings reveal complicated and ecologically differentiated environment-life processes, indicating the importance of high-resolution temporal data for dissecting out ecological responses to major environmental changes.

The Eocene–Oligocene transition (EOT), ca. 34.1–33.7 million years ago (Ma), was a turning point in Earth's history when the climate moved from a warmhouse to an icehouse state[1,2]. The climate shift was marked by terrestrial and ocean cooling[3–5] with large ice sheets emerging on

Antarctica during the early Oligocene[6]. The trigger of this climate transition has been identified as the drawdown of atmospheric $p$CO$_2$[7] through weathering of continental silicates[8], enhanced oceanic productivity[9], and/or changes in ocean circulation controlling

---

[1]State Key Laboratory of Critical Earth Material Cycling and Mineral Deposits, School of Earth Sciences and Engineering, Nanjing University, Nanjing, China. [2]Frontiers Science Center for Critical Earth Material Cycling, Nanjing University, Nanjing, China. [3]National Key Laboratory for Novel Software Technology, Nanjing University, Nanjing, China. [4]School of Artificial Intelligence, Nanjing University, Nanjing, China. [5]School of Resources and Environmental Engineering, Hefei University of Technology, Hefei, China. [6]School of Earth and Planetary Sciences, East China University of Technology, Nanchang, China. [7]Department of Earth Sciences, University College London, London, UK. [8]Department of Earth Sciences, University of Zaragoza, Zaragoza, Spain. [9]School of Earth Sciences, Wills Memorial Building, University of Bristol, Bristol, UK. [10]Department of Earth Sciences, University of California, Riverside, CA, USA. [11]These authors contributed equally: Zhengbo Lu, Ke Xue, Yiying Deng. ✉e-mail: jxfan@nju.edu.cn; qianc@nju.edu.cn; ykshi@nju.edu.cn

poleward heat transport and carbon burial[10]. Antarctic glaciation caused significant (-55 m) glacio-eustatic sea-level fall in the early Oligocene, coincident with the positive $\delta^{18}O$ shift, Oi-1 at -33.65 Ma[11]. The initiation of Oligocene glaciation was also accompanied by an ocean-wide positive $\delta^{13}C$ anomaly of up to 1.0‰[6] attributed to sea-level fall and reorganisation of the carbon cycle in response to a rapid drawdown of atmospheric $pCO_2$[12]. Moreover, distinct geological events, such as bolide impacts[13] and volcanic activity in large igneous provinces (LIPs)[14], have been documented in the late Eocene and early Oligocene. The eruption of flood basalts related to the Afar-Arabian LIP from -32 to 28 Ma[14–16] might have been associated with contemporary environmental changes (e.g., global cooling and aridity), resulting in extinctions of land mammals[14].

Biosphere disturbances during the EOT have been reported[1], especially for foraminifera[17], which were widespread in Cenozoic oceans and have been used extensively for stratigraphic correlation and palaeoclimatic reconstruction. Foraminifera are single-celled protists that can be separated into three groups that are generally distinct in their life-history strategy, morphology, and ecology, namely the planktonic foraminifera (PF), larger benthic foraminifera (LBF) and small benthic foraminifera (SBF). PF are free-floating and primarily inhabit open ocean waters, with their distribution strongly influenced by sea surface temperature, salinity, oxygen levels, nutrient availability, and light intensity. These factors not only affect their survival and vertical distribution but also their symbiotic relationships with photosynthetic organisms. SBF, on the other hand, are more ecologically versatile, occupying a broad range of depths from shallow to deep marine settings. Their survival is primarily governed by dissolved oxygen levels and food availability, and historical fluctuations in these parameters have been linked to major extinction events in foraminiferal history. In contrast, LBF are predominantly found in warm, shallow tropical to subtropical marine environments and are characterised by complex internal structures. Their ecological range is narrower than that of SBF, and it is controlled by light intensity, nutrient availability, and salinity because of their reliance on phototrophic symbionts (See Supplementary Note 1). They are abundant, geographically widespread and sensitive to environmental changes[18,19]. During the EOT, planktonic and benthic foraminifera suffered a richness decline for more than 6 Myr[20,21]. Deep-sea benthic foraminifera underwent a turnover during the late Eocene to early Oligocene[22–24]. The EOT also witnessed the extinction of some long-ranging and widespread families of PF and LBF, such as Hantkeninidae and Discocyclinidae[25–28]. The triggers of these changes have been controversially related to progressive cooling, sea-level changes associated with glaciation, and changes in nutrient supply[22–28].

Although the effects of the EOT on marine life have been recognised for over 80 years[29], the timing and pattern of biotic changes, particularly those linked to different environmental factors, remain debated. While previous large-scale studies and compilations have mostly focused on planktonic foraminifera[30–32], including analyses of their ecologically driven diversity dynamics[33] and comparisons with other microfossil groups such as diatoms and calcareous nannofossils[32,34], comparative studies involving benthic foraminifera—descendants of the same ancestral lineage but with distinct ecological characteristics—remain limited, largely reflecting field-wide challenges inherent to large-scale synthesis (e.g., age calibration difficulties, taxonomic inconsistencies, and high levels of endemism). Furthermore, most richness reconstructions rely on binning schemes with uneven or highly variable temporal resolutions[30,35,36], though a few studies have applied unbinned methods[33]. Consequently, our understanding of how organisms with diverse ecological behaviours respond to the substantial physical and chemical environmental changes across the entire ocean is still limited.

Here, we compiled global foraminifera data, including both planktonic (macroperforate and microperforate) and small and larger benthic foraminifera. An AI-inspired metaheuristic variant of the Constrained Optimization (CONOP) method[37–40]—CONOP.EA, which incorporates an evolutionary algorithm—is applied to this comprehensive dataset. This method was used to automatically correlate stratigraphic sections/sites of the Eocene and Oligocene, enabling the construction of a high-resolution foraminiferal richness curve spanning 48 Ma to 20 Ma. This high-resolution pattern is subsequently used to identify major biotic changes in foraminifera during the Eocene and Oligocene, and especially across the EOT. Richness changes across the different groups of foraminifera are examined to throw light on the major biotic changes during the Eocene and Oligocene, and to quantitatively clarify the proposed relationship between foraminiferal richness and environmental changes, encompassing carbon isotopes, sea-surface temperature, deep-ocean temperature and eustatic sea level.

## Results and discussion
### Reconstructing high-resolution chronology
Achieving high-resolution chronology in deep time (e.g., EOT) requires exploiting the full information behind first and last appearances of multiple fossil species[41], compared to traditional zonal biostratigraphy, which prioritises the most common and well-studied species and uses only ~1–10% of the fossil record. High-resolution chronology thus necessitates extensive, well-sampled, and multi-sourced stratigraphic datasets, coupled with quantitative stratigraphic methods, such as CONOP which integrates patchy local bioevents into an optimal composite sequence that best matches the fossil ranges observed in numerous local sections/drill cores[37–42]. This method aims to estimate the most complete range of species among sections/cores in the study dataset, thereby providing an imputed, algorithm-based temporal framework for high-resolution analysis of richness patterns in life-evolution studies[37,43,44].

The CONOP.EA algorithm used in this study applies a procedure, inspired by Darwin's theory of evolution, to iteratively improve a set (i.e., population) of composite sequences via procedures that resembles or mimics mutation, recombination, and natural selection (Fig. 1). The composite sequence, which is a certain sequence of stratigraphic bioevents (i.e., first and last appearance datums) of all involved taxa, can be considered as a half DNA sequence that represents the genome of an individual organism. The CONOP.EA program maintains a population of individual sequences of constant size, e.g., 24, the number of CPU cores used in a calculation. Starting from an initial population of diverse sequences, CONOP.EA produces offspring sequences by recombining randomly selected segments of two-parent sequences (i.e., recombination) and randomly moving the positions of some datums in the sequence (i.e., mutation). The generated offspring should obey basic stratigraphic constraints, e.g., a taxon's last appearance datum must be younger than its own first appearance datum, and observed coexistences among different species should still exist, assuming that the observed local ranges cannot be longer than the true global ranges[41]. Those offspring with problematic or fatal mutations are excluded. Then, the program determines which of the remaining offspring, if any, provides a better fit (i.e., lower penalty) for the dataset as a whole. If found, the optimised sequence is included in the population to replace the current least optimised sequence so that the population size is kept constant. Through this selection procedure, the offspring sequences with problematic or fatal mutations and larger penalties do not survive and reproduce. Because the optimised sequences possessing certain datum variations tend to survive subsequent optimality evaluations and then generate new, further optimised sequences more frequently, the population evolves from one iteration (=generation) to another until a stable convergence is achieved. The optimal datum sequence in the final population is then selected for subsequent richness analysis.

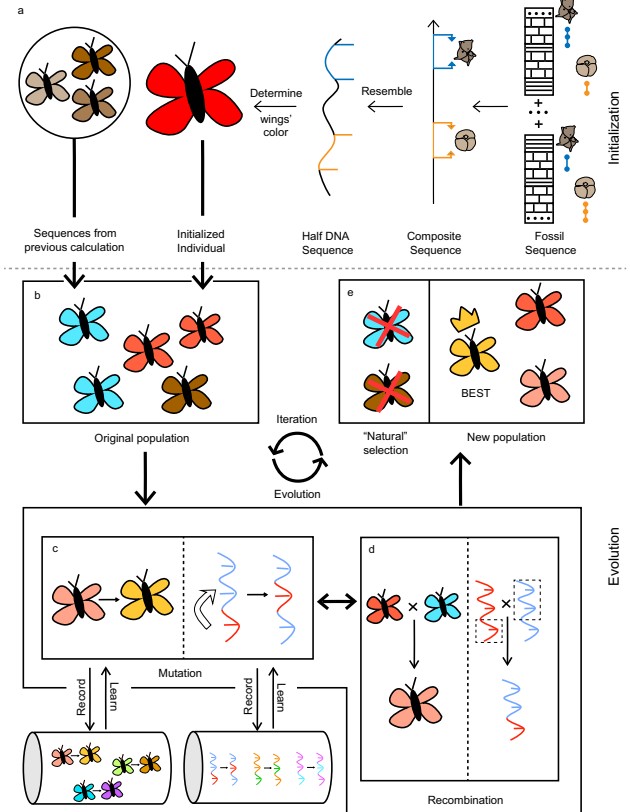

**Fig. 1 | Overview of the CONOP.EA procedure. a** Initialisation process. Consider a nucleic acid, which determines the wing colour of butterfly individuals, as representing a certain sequence of stratigraphic bioevents. A bioevent, i.e., first or last appearance of a taxon, can be considered as a nucleotide. A composite sequence for a given stratigraphic dataset resembles a DNA sequence composed of nucleotides. **b** Original population. The original population should contain diverse sequences, sometimes varying greatly. New individuals (equal to composite sequences) may be generated in two ways, by combining bioevents from sections or referring to best composite sequences from previous calculations. **c** Mutation. This process can occur randomly in two ways: change the positions of two neighboring stratigraphic datums in the sequence (i.e., small mutation) or move stratigraphic datum 1 to the position of stratigraphic datum 2, with the datums between datums 1 and 2 all moving one place in time (i.e., large mutation). If the generated sequence offends the constraint rules (See description in the text), it is refused. Every mutation step that improves the sequence is recorded and learned by subsequent mutations to generate stronger offspring. **d** Recombination. Select two parent sequences, determine the segment for crossover, and replace the segment in sequence 1 with the segment in sequence 2. The generated sequence also must obey the constraint rules. **e** Natural selection. First, replace worse individuals (larger penalty) in the population with newly generated, better ones (less penalty), which means the population is only composed of the currently best individuals. Second, the optimised individuals in the population, which contain certain datum variations, tend to be selected more frequently as parents to reproduce more offspring than individuals with other less successful variations. This procedure from steps c to e is repeated while the population evolves from one generation to the next until convergence is reached.

It is noteworthy that optimisation of a set of stratigraphic datum sequences is often very complex, in that many local optima exist[45,46]. To enhance the search ability of CONOP.EA, a series of components was designed, guided by recent theoretical advances in evolutionary learning[46]. The fact that CONOP.EA maintains a diverse population of sequences (instead of only one sequence) and produces offspring sequences by recombination provides the current program with an improved ability to escape from local optima than the original CONOP[47] or CONOP.SAGA[37] procedures (Fig. S1), thus rendering it more likely to find the globally optimised solution. Furthermore,

CONOP.EA utilises domain knowledge as well as learning from the information accumulated during the run to accelerate convergence. For example, to generate the initialising individual in the original population, the study sections need to be ranked by numbers of common species weighted by numbers of occurrences of each species. From this information, the first section in the rank is selected to generate an individual sequence by adding species from other sections to it one by one. It should be noted that integrating domain knowledge in the process of utilising data has been a promising direction in AI (e.g., abductive learning[48]).

The CONOP.EA output is a high-resolution, event-dense composite sequence that aligns stratigraphic events in the optimal order consistent with local observations. By integrating fossil and stratigraphic data from multiple sections, CONOP.EA constructs a coherent, unified chronological framework that captures the most complete possible range of species across the dataset. This synthesis enhances temporal resolution, allowing fine-scale detection of biodiversity dynamics, evolutionary rates, and turnover patterns[37,43,44]. Compared to traditional methods, this approach offers a more continuous and globally consistent timeline, making it an effective framework for resolving critical intervals such as the Eocene−Oligocene transition.

The calibrated CONOP-derived composite comprises 962 discrete temporal levels, each representing a cluster of imputed contemporaneous bioevents (i.e., first and last appearance datums). This composite spans an interval from 48 Ma to 20 Ma, yielding an imputed temporal resolution of approximately 29.11 Kyr (calculated as 28 Myr span divided by 962 levels), to which the relative ordering of bioevents can be resolved. It should be noted that this represents an imputed temporal resolution, which is conceptually distinct from the absolute age resolution obtained through isotope dating techniques.

## Fine-scale diversity trajectory

Foraminifera can document wider changes in marine biodiversity because their varied ecologies mean they represent most biotopes. From 48 Ma to 20 Ma, foraminiferal richness shows notable fluctuations, characterised by two phases of increase and two of decline (Fig. 2a–c). In addition, each foraminiferal group exhibits a distinct richness trajectory (Fig. 3a).

Foraminifera display an increasing species richness pattern in the early and middle Lutetian (Event No.1 in Fig. 2), followed by the Bartonian richness decline (BRD, Event No.2 in Fig. 2), including a long-term decrease from the late Lutetian to earliest Priabonian (Fig. 2a–c). The decrease was generally gradual with low proportional origination rate (-0.2/Lmyr on average, where /Lmyr denotes species per lineage per million years; Fig. 2d), despite notable peaks at the beginning and end in proportional extinction rate (Fig. 2e). The BRD was followed by a rebound in richness, the early-Priabonian radiation (EPR) (Event No. 3 in Fig. 2). During the early Priabonian, species richness nearly doubled, and the proportional origination rate reached two peaks of -0.9 and -0.8/Lmyr and maintained a consistently high level (Fig. 2d).

The Eocene−Oligocene richness crisis is evident as a long-term richness reduction from the late Priabonian to the earliest Chattian (from 35.06 ± 0.41 Ma to 25.59 ± 0.66 Ma, reported as the midpoint ± half-width of the bootstrap 95% confidence interval; Fig. 2b). This richness decline can be divided into two phases, a long-term and progressive Eocene−Oligocene extinction (EOE, Event No. 4 in Fig. 2), extending from the late Priabonian to early Rupelian (from 35.06 ± 0.41 Ma to 32.01 ± 0.55 Ma) and a subsequent, sustained Rupelian richness decline (RRD, Event No. 5 in Fig. 2, from 30.96 ± 0.58 Ma to 25.59 ± 0.66 Ma). The EOE was characterised by high proportional extinction (-0.3/Lmyr on average) and low proportional origination (-0.1/Lmyr on average) rates and thus represents a straightforward richness decrease. The RRD was also characterised by high proportional extinction (-0.2/Lmyr on average) and low

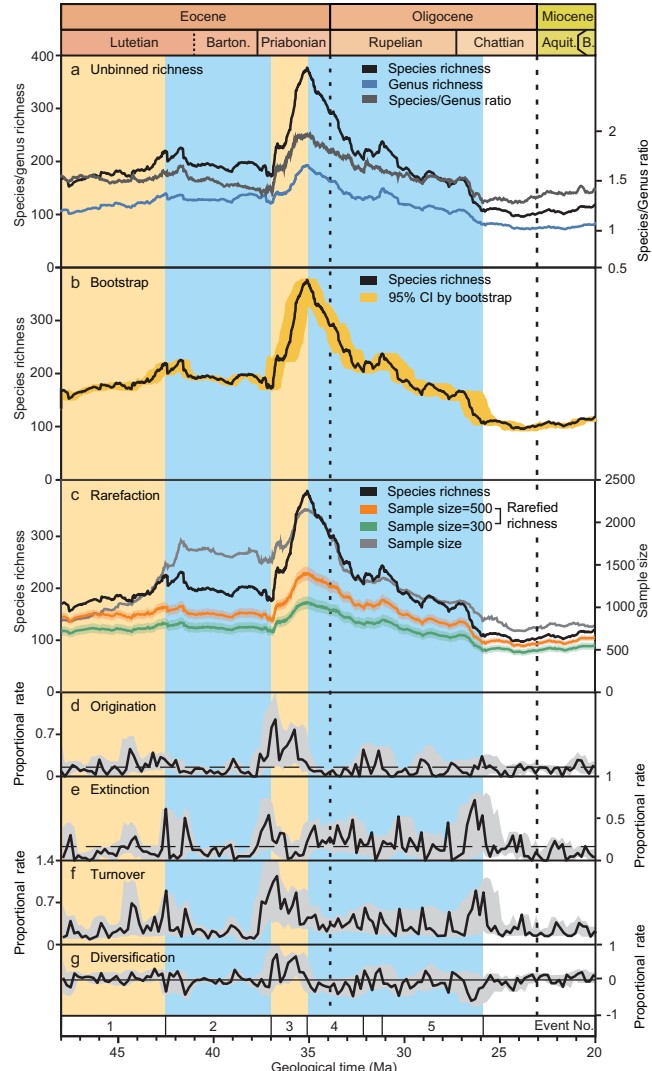

**Fig. 2 | General trajectories of foraminiferal species richness. a** Species richness, genus richness and species/genus ratio. **b** Species richness (solid line shows the point estimate from the original data) and 95% confidence intervals following 1000 bootstrap iterations. **c** Rarefied richness for sample sizes of 300 and 500, respectively (solid lines show the mean across 1000 rarefaction iterations and shading indicates 95% confidence intervals). **d** Proportional origination rate (solid line shows the point estimate from the original data) and 95% confidence intervals following 1000 bootstrap iterations. Dashed line shows the average. **e** Proportional extinction rate (solid line shows the point estimate from the original data) and 95% confidence intervals following 1000 bootstrap iterations. Dashed line shows the average. **f** Proportional turnover rate (solid line shows the point estimate from the original data) and 95% confidence intervals following 1000 bootstrap iterations. **g** Proportional diversification rate (solid line shows the point estimate from the original data) and 95% confidence intervals following 1000 bootstrap iterations. The time scale is standardised to GTS 2020[64,65]. 1, Lutetian richness increase; 2, Bartonian richness decline (BRD); 3, early-Priabonian radiation (EPR); 4–5: Eocene−Oligocene richness crisis, including Eocene−Oligocene extinction (EOE) and Rupelian richness decline (RRD). Barton., Bartonian; Aquit., Aquitanian; B., Burdigalian.

proportional origination (~0.1/Lmyr on average) rates, which fluctuated, thus accounting for the interval's richness volatility. The Eocene−Oligocene richness crisis ended with two high proportional extinction peaks (~0.7/Lmyr and ~0.5/Lmyr) in the earliest Chattian.

In the Chattian to early Aquitanian, foraminiferal species richness remained low, with generally low proportional origination and extinction rates (Fig. 2a, d, e). Nevertheless, an increase in species/

genus ratio and proportional origination rate in the Aquitanian, possibly implies the start of an early Miocene recovery (Fig. 2a, d).

The species/genus ratio generally shows a comparable trend to that of the richness change (Fig. 2a), implying that species richness varied while genus richness remained relatively stable. The average value of the ratio is ~1.6 throughout the study interval.

Long-term trends in foraminiferal richness have already been established, but unlike previous studies, our record is at higher resolution (~29 Kyr as imputed resolution) covering all three groups of foraminifera. Our PF and LBF richness curves are comparable to those of previous studies[30,31,49]. The strong correlation between our PF and LBF richness time series ($\rho = 0.87$; Fig. 3 and Table S1) provides evidence for similarity between ecological trends in planktonic and larger benthic foraminifera over the time interval[50]. The LBF richness curve of Whidden and Jones[49] exhibited a long-term decline from the middle Lutetian to the end of the Rupelian, similar to that of our study (Fig. S2a). The only major difference between these two studies is the more obvious early Rupelian richness decline documented in our results, whereas the earlier study documented a decrease in the late Rupelian. The PF species richness curve of Fenton and Woodhouse et al.[30] resembles our curve in most features, especially the progressive and gradual richness decline in the Bartonian and Priabonian and the rapid, earliest Rupelian richness decrease (Fig. S2b). The PF species richness curve of Fraass et al.[31] presented a similar gradual reduction in richness across the Eocene/Oligocene boundary (Fig. S2c). Furthermore, both curves present a slow but gradual richness increase in the Aquitanian, indicating PF recovery after the EOE.

Our study reveals notable differences in the richness patterns of SBF, compared with PF and LBF (Fig. 3a). Alegret et al.[22] also observed a comparable general pattern of SBF diversity, with a moderate increase in the early Eocene, followed by a decrease during cooling periods, besides a diversity increase in the Pacific Ocean, attesting the establishment of the latitudinal gradient of SBF richness during the early Lutetian. Our results show that a slow richness increase occurred in the early Lutetian, ending with a small drop in richness in the late Lutetian. After a long interval of stable richness values in the Bartonian, SBF were characterised by a quick (~1.9 Myr) richness increase (i.e., EPR), owing to an interval of high proportional origination rates (Fig. S3a, d). This similar increase can also be observed in the Pacific Ocean[22]. While both PF and LBF exhibited rapid richness declines near the Eocene/Oligocene boundary, SBF showed a ~9-Myr-long deterioration in richness from the late Priabonian to the end of the Rupelian. This decrease was characterised by highly fluctuating proportional origination and extinction rates, with the proportional extinction rate generally higher than the proportional origination rate (Fig. S3a, d).

## Mechanisms of long-term richness changes
Foraminifera, especially those inhabiting different depths in the water column, appear to show differentiated diversity changes corresponding to related environmental changes[21,22,31,33,51]. This complex differentiation mechanism involving multiple environmental factors has not been fully elucidated yet in deep-time studies. Our high-resolution foraminiferal time series richness data covering 28 Myr offers the chance to test the relationships between some combinations of ecologically long-term environmental factors and foraminiferal species dynamics (Fig. 3).

Over the study interval, three groups of foraminifera show various and close correlation with environmental factors or proxies before and/or after detrending. Both PF and LBF richness curves are strongly correlated with sea level (both $\rho = 0.74$ and 0.70, respectively) and sea-surface temperature ($\rho = 0.85$ and 0.74, respectively) (Table 1). Only SBF show significant correlation with carbon isotopes ($P < 0.05$), though the correlation coefficient is weak ($\rho = 0.23$). After detrending, this correlation remains significant, with $\rho = 0.19$ and $P < 0.05$. SBF also exhibit a moderate to strong correlation to deep-sea temperature

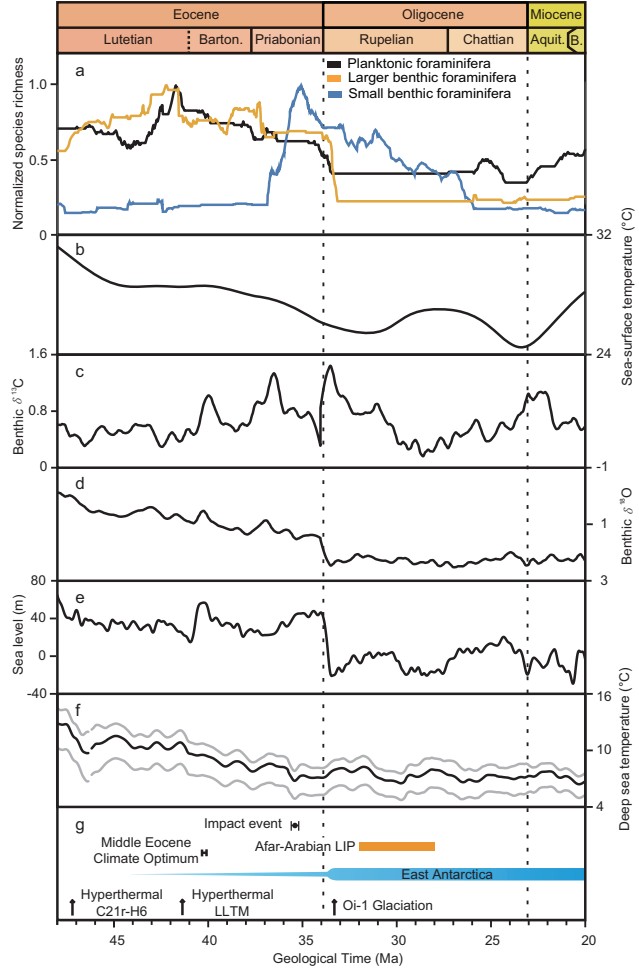

**Fig. 3 | Comparison of species richness curves and environmental proxies.**
**a** Max-Min normalised species richness of three groups of foraminifera. Planktonic foraminifera (black), larger benthic foraminifera (orange) and small benthic foraminifera (blue). **b** Sea-surface temperature[67]. **c** Benthic $\delta^{13}C^2$. **d** Benthic $\delta^{18}O^2$. **e** Eustatic sea level[11]. **f** Deep-ocean temperature[66]. Grey lines represent 90% confidence intervals. **g** Distinct geological and climatic events. The time scale for sea level, sea-surface and deep-sea temperature is modified to the GTS 2020[64,65]. The age models for oxygen and carbon isotopes are based on Westerhold et al.[2] LIP, Large Igneous Province; LLTM, Late Lutetian Thermal Maximum.

($\rho = -0.62$) in the detrended time series, but only weak correlations to those environmental factors or proxies prior to detrending.

Correlations before and after detrending reveal potential secular and short-term influences of multiple environmental factors on richness changes[52]. Our results indicate that sea-surface temperature and eustatic sea-level changes correlate with fluctuations in richness for planktonic[33] and larger benthic foraminifera over the entire 28-Myr study interval[27]. Global cooling can lead to a decrease in planktonic and larger benthic foraminifera species richness by reducing diversity of warm-water taxa[50] that thrived on the warmhouse planet. Sea-level change can reduce habitat area, intensify biotic competition and ultimately cause richness declines[19,25]. An explanation for the negative correlation between SBF and deep-sea temperature after detrending (Table 1) is increased food supply through enhanced phytodetritus deposition. With ocean cooling, a greater proportion of the organic carbon fixed in the surface ocean would have reached the sea floor due to the temperature dependency of the ocean's biological carbon pump[53]. This model is consistent with our finding of positive correlation between SBF and carbon isotopic values (Table 1).

## Diversification and extinction events

In addition to the long-term environment-life relationship, certain rapid environmental shifts can sometimes induce transient biological changes[54]. The high-resolution time series richness data provide finer correlations to the occurrence of the environmental changes, allowing for a more precise elucidation of the environmental catalysts underpinning the three major biological changes during the EOT, including the early Priabonian radiation (EPR, Event No. 3 in Fig. 2), Eocene-Oligocene extinction (EOE, Event No.4 in Fig. 2) and Rupelian richness decline (RRD, Event No. 5 in Fig. 2).

The Priabonian was a time of major changes in marine ecosystems, with phytoplankton blooms in the oceans[23,55] and a high export productivity event (~37 Ma)[9]. Marine diatoms, which account for half of organic carbon burial in the ocean, diversified in pulses after ~38 Ma, coupled with global cooling[55,56]. This diversification, combined with cooling, would lead to increased export of organic matter and changes in the amount of organic matter reaching the seafloor[9,24,53,56], perhaps accounting for the radiation of deep-sea SBF. However, the early Priabonian radiation does not appear to be reflected in the PF and LBF results, as their richness remains stable. The observed stasis in richness is consistent with the distinct ecological characteristics of PF and LBF. In general, PF and LBF can access food in the euphotic zone and also rely on photosymbiotic algae. The blooming of diatoms, to some extent, results in rapid uptake of surface nutrients and the transport of organic carbon to the sea floor (i.e. biological pump), leading to differences in nutrient content between shallow environments, surface waters, and deep seafloor regions. Consequently, during the time characterised by increasing nutrient export, the richness of PF and LBF shows a comparatively small change when compared to SBF because of their distinct ecological features.

Foraminiferal changes across the EOT have been linked to abiotic shifts[22-33]. Previous studies have revealed that oxygen isotopic values from planktonic foraminiferal tests display a positive excursion roughly correlated with the turnover and extinction of PF and LBF[27,28]. This association suggests that global cooling might have driven these foraminiferal events[28]. Additionally, sea level dropped precipitously in the early Oligocene (~33.65 Ma), caused by major glaciation (Oi-1)[11]. Our results indicate that sudden PF and LBF richness decreases near the Eocene/Oligocene boundary may be related to the combination of sea-surface temperature decrease and eustatic sea-level fall[27,28] (Fig. 3).

SBF have different ecological features and therefore show a different extinction pattern across the Eocene/Oligocene boundary, compared with PF and LBF. After the early Priabonian radiation, SBF experienced a ~9-Myr, two-phase richness decline: the Eocene-Oligocene extinction and a Rupelian richness decrease. The first phase (from $35.06 \pm 0.41$ Ma to $33.88 \pm 0.52$ Ma) started in the late Priabonian and ended before the extinction of PF and LBF across the Eocene/Oligocene boundary. The richness reduction in SBF was accompanied by a series of complex biotic and abiotic changes. Major primary producers in the ocean underwent changes that could have impacted the food sources and composition reaching the seafloor[23,55,57]. High-latitude food supply became seasonal[24], while oligotrophic conditions became more pronounced[22]. Perturbations in nitrogen and sulfur isotopes, associated with microbial activity, were triggered by cooling and sea-level variations[58], suggesting changes in the rate of remineralisation, ocean oxygenation and the size of the dissolved organic oceanic reservoir[22]. All these changes probably led to the decrease in the SBF richness, particularly the previously proposed high-latitude diversity reduction[24].

Links between bolide impacts (Popigai and Chesapeake) and the Eocene−Oligocene extinction have been discussed ever since the discovery of physical evidence, including at least two horizons associated with bolide impacts from sections and drill cores[13]. The timing of these major impact events is estimated to be in the mid-Priabonian, $35.5 \pm 0.2$ Ma[13], approximately 1.6 Myr before the rapid growth of the

**Table 1 | Spearman's rank correlation between species richness and climate proxies**

| | Sea level[11] | | T$_{deep\ ocean}$[66] | | δ$^{13}$C$_{benthic}$[2] | | SST[67] | |
|---|---|---|---|---|---|---|---|---|
| | ρ (95% CI) | P | ρ (95% CI) | P | ρ (95% CI) | P | ρ (95% CI) | P |
| Raw | | | | | | | | |
| F | 0.34 (0.18, 0.48) | $5.04 \times 10^{-5}$ | 0.21 (0.04, 0.37) | 0.011 | 0.28 (0.11, 0.44) | $7.81 \times 10^{-4}$ | 0.11 (−0.08, 0.30) | 0.184 |
| PF | 0.74 (0.68, 0.79) | $7.17 \times 10^{-26}$ | 0.68 (0.60, 0.73) | $3.32 \times 10^{-20}$ | −0.11 (−0.28, 0.07) | 0.203 | 0.85 (0.81, 0.88) | $2.65 \times 10^{-40}$ |
| LBF | 0.70 (0.64, 0.75) | $2.9 \times 10^{-22}$ | 0.67 (0.58, 0.73) | $3.02 \times 10^{-19}$ | −0.01 (−0.18, 0.16) | 0.928 | 0.74 (0.68, 0.79) | $5.09 \times 10^{-26}$ |
| SBF | −0.11 (−0.30, 0.09) | 0.214 | −0.23 (−0.40, −0.06) | 0.005 | 0.23 (0.06, 0.39) | 0.006 | −0.33 (−0.50, −0.16) | $5.94 \times 10^{-5}$ |
| Detrended | | | | | | | | |
| F | −0.16 (−0.36, 0.03) | 0.052 | −0.61 (−0.71, −0.48) | $<2.2 \times 10^{-308}$ | 0.24 (0.07, 0.39) | 0.005 | −0.45 (−0.59, −0.29) | $2.62 \times 10^{-8}$ |
| PF | 0.53 (0.38, 0.65) | $<2.2 \times 10^{-308}$ | 0.10 (−0.06, 0.25) | 0.241 | −0.01 (−0.19, 0.16) | 0.884 | 0.56 (0.41, 0.67) | $<2.2 \times 10^{-308}$ |
| LBF | 0.55 (0.42, 0.66) | $<2.2 \times 10^{-308}$ | −0.11 (−0.29, 0.09) | 0.208 | 0.21 (0.03, 0.38) | 0.015 | 0.14 (−0.05, 0.33) | 0.088 |
| SBF | −0.23 (−0.42, −0.03) | 0.007 | −0.62 (−0.72, −0.50) | $<2.2 \times 10^{-308}$ | 0.19 (0.01, 0.36) | 0.026 | −0.50 (−0.64, −0.33) | $6.35 \times 10^{-10}$ |

ρ is Spearman's rank correlation coefficient; P values are two-tailed; 95% CIs are percentile bootstrap intervals (2.5th–97.5th percentiles; Bootstrap iterations = 10,000). Very small P values are reported as $P < 2.2 \times 10^{-308}$. T$_{deep\ ocean}$, deep-ocean temperature; δ$^{13}$C$_{benthic}$, benthic foraminiferal δ$^{13}$C; SST, sea-surface temperature.

Antarctic ice sheet, sea-level drop and extinction of PF or LBF (Fig. 3)[59]. The initiation of the SBF richness decline started at 35.06 ± 0.41 Ma, with a notably high extinction rate (Fig. S3a, d). Nevertheless, there is no published evidence of any effect of the bolide impact on primary producers or on species that occupy a crucial position at the base of marine food chains, such as calcareous nannofossils, diatoms, PF, and LBF. Clear responses of PF and LBF are not observed to any bolide impact in this study. Additionally, there is no evidence that consequent physical or chemical changes, e.g., oceanic redox state, during the brief period under consideration, directly impacted the survival of SBF in the deep ocean[60]. In conclusion, there appears to be no relationship between the Priabonian impact events and foraminiferal diversity change.

After the E/O boundary, SBF richness experienced a stagnant phase for ~1.1 Myr, followed by a ~7-Myr lowering with highly fluctuating proportional origination and extinction rates, named the Rupelian richness decline (from 32.78 ± 0.57 Ma to 25.74 ± 0.66 Ma; Fig. 3a, Fig. S3a, d). One factor that might have contributed to this decline is the contemporaneous Afar-Arabian LIP event (Fig. 3g). LIP eruptions have been regarded as potential causes of many extinction events in geologic history[15]. The flood basalt eruptions related to Afar-Arabian LIP activity have been documented to be ~32–28 Ma in age[14-16]. A recent study has suggested that LIP activity can lead to significant expansion of oceanic anoxia and euxinia (covering an area of seafloor ~1–2 orders of magnitude greater than today)[14-16], which could affect foraminiferal assemblages in the ocean, especially on the sea floor, possibly indicating an association between the Afar-Arabian LIP activity and the SBF crisis.

## Summary

Although long-term environmental perturbations influence biological evolution, testing deep-time environment-life hypotheses at a level of detail approaching that of long-term ecological analyses (~10,000 years) is challenging[37] through the lack of high-resolution richness data (e.g., Cenozoic) or high-resolution environmental proxies (e.g., Paleozoic or Mesozoic). In this study, we designed an AI-inspired metaheuristics algorithm to construct a high-resolution global foraminifera species richness history across the Eocene-Oligocene transition with an imputed resolution of ~29 Kyr. These high-resolution time series richness data offer the chance to test the relationships between long-term environmental factors and foraminiferal species dynamics. Our results indicate that sea-surface temperature and eustatic sea-level changes strongly correlate with fluctuations in richness for planktonic and shallow-water larger benthic foraminifera over the entire 28-Myr study interval, while deeper-water small benthic foraminifera show significant correlation with carbon isotopes for this interval. In finer detail, planktonic and larger benthic foraminifera show steady diversity levels in the early phases of the transition in the latest Eocene after a long-term decline, while the small benthic foraminifera exhibit a notable radiation and subsequent richness decline over the same interval. In the earliest Oligocene, the planktonic and larger foraminifera suffer major species losses coincident with the first permanent ice sheets formed in Antarctica, while small foraminiferal diversity held steady, followed by an accelerating decline as the early Oligocene proceeded.

This study provides a high-resolution perspective on the links between environmental change and biodiversity history over a long-time scale (~28 Myr) in deep time. This improved resolution reveals that foraminifera living in different marine habitats had distinct evolutionary histories, dominated by various environmental factors influencing different water depths or habitats. These findings suggest that environment-life relationships during the EOT were much more complex and ecologically nuanced than previously reported, and that they can be tested with high-quality data. This necessitates a refocus of major biotic events in deep time, and high-resolution fossil data and environmental proxies are crucial for this re-evaluation.

## Methods

### Data overview

Our assembled global foraminifera dataset (raw dataset) comprises 13,138 local bioevents records (i.e., first and last appearance records) and ~60,000 occurrences of 2988 taxonomic units from 163 published stratigraphic sections (Supplementary Data 1), encompassing both calcareous and agglutinated foraminifera. After the cleaning process (See Supplementary Notes 2–4), the final dataset comprises approximately 40,000 fossil occurrences of 1269 species derived from 161 stratigraphic records (drill cores and outcrops) (Fig. S4; Supplementary Data 2).

The dataset includes stratigraphic data for three foraminiferal groups: 277 PF species, 340 LBF species and 652 SBF species. Of the 161 stratigraphic records (drill cores and outcrops), 102 yield one foraminiferal group, 59 yield at least two groups, and 16 include all three groups, covering major ocean basins, including the Atlantic and Indian Oceans, as well as key marginal seas such as the Mediterranean, Caribbean, and Gulf of Mexico (Fig. S4).

Occurrence data were extracted from original biostratigraphic distribution records (depth-versus-taxon) reported in peer-reviewed journal articles and scientific reports, including 10 continental drill cores, 52 ocean drill cores (DSDP, ODP and others), and 99 outcrops. In addition, 18 magnetochrons from five GSSP or GSSP candidate sections (Massignano, Oyambre, Agost, Varignano, and Alano) were incorporated into the dataset to enhance stratigraphic correlation and support robust age calibration.

In terms of depositional environments, 77 outcrops/cores (~48%) are assigned to coastal settings, e.g., shallow marine, inner to middle shelf, neritic, and lagoon, reflecting deposition within the photic zone and proximal to land (Supplementary Data 2). Sixty-nine outcrops/cores (~43%) were attributed to deep marine settings, i.e., bathyal, abyssal, continental slope, and pelagic environments, representing deposition below the photic zone and distal from siliciclastic input. The remaining 15 outcrops/cores (~9%) exhibited mixed shallow–deep characteristics, indicating transitional depositional settings along the shelf-to-basin gradient.

## Data preparation and standardisation

The dataset analysed here was manually digitised by the OneStratigraphy team between May 2020 and October 2021. It covers a wide range of published literature and therefore incorporates records documented under diverse reporting styles, formats, and taxonomic conventions. The corresponding reference sources are traceable in the OneStratigraphy Database through the bibliographic information provided in Supplementary Data 2. Prior to analysis, the dataset was first cleansed and standardised by excluding non-foraminifera fossils, addressing open nomenclature, correcting typographical errors, etc. (See Supplementary Notes 2). All taxonomy was thoroughly examined through use of taxonomic atlases[61,62], foraminiferal databases (Mikrotax and WoRMS), and related taxonomic references. These opinions were finally verified and resolved by a group of foraminiferal taxonomic experts for correctness and consistency: Bridget Wade (PF), Laia Alegret (SBF), Qinghai Zhang (LBF and SBF), and Peiyue Fang (PF, LBF and SBF).

To ensure that our subsequent analyses are based on data with sufficient temporal resolution and reliability, we first evaluated the stratigraphic resolution of PF, SBF, and LBF. A pre-estimated result shows that PF exhibited the highest resolution (~0.30 levels per section), followed by SBF at ~0.24 levels per section, while LBF demonstrated the lowest resolution (~0.08 levels per section). This resolution is estimated from the average number of independent bioevents per section—that is, fossil occurrences that do not coincide with any other taxon at the same stratigraphic level. A higher average indicates finer temporal resolution, because more unique events allow us to distinguish shorter time intervals, and vice versa. To minimise sampling bias, we included in our analysis only stratigraphic sections containing more than 30 biostratigraphic events, thereby ensuring sufficient data density.

## CONOP.EA: An evolutionary algorithm for global correlation

The original CONOP program was first developed by Peter Sadler to address the problem of integrating patchy local biostratigraphic events into a globally consistent temporal ordering[39,41]. It was designed on a simulated annealing algorithm and did not support either parallel computing or high-performance computing. Fan et al.[37] developed an enhanced program, CONOP.SAGA, by combining simulated annealing and a genetic algorithm to overcome these limitations. CONOP.SAGA was largely based on the original CONOP program designed by Sadler et al.[41] It followed the same procedure as in CONOP to generate the initialising sequence, i.e., randomly placing the first appearance datums to positions in the first half of the sequence and the last appearance datums to positions in the last half of the sequence. It contains only the mutation procedure of the evolutionary algorithm and records those mutation operations that improved the sequence in previous iterations. These recorded operations were collected to construct a better sequence while all the CPU cores communicated after finishing a fixed number of iterations (e.g., 10,000 iterations). This better solution was then shared by each CPU core as the starting point for the next round of iterations.

Here, we introduce a new evolutionary-algorithm-based implementation, CONOP.EA, which further enhances both computational performance and optimisation capability. It is designed to handle datasets of more than 10,000 species, which is difficult for CONOP and CONOP.SAGA, as the cost in terms of supercomputer time is enormous and unacceptable. The EA in its name denotes the integration of a complete evolutionary algorithm into the program.

CONOP.EA has many distinct features, but two are most important. First, it utilises a more complicated evolutionary algorithm (Fig. 1) than CONOP.SAGA. In each calculation, CONOP.EA generates an initial population of many sequences (Fig. 1b), and the population evolves from one generation to another by mutation, recombination and natural selection (Fig. 1c–e). Second, CONOP.EA can start from a dataset of sections, similar to CONOP or CONOP.SAGA, or from a set of sequences constructed before, e.g., result sequences from CONOP or CONOP.SAGA calculations. This allows CONOP.EA to compare the differences among previous results and recombine those that are distinct to generate offspring sequences that possess a better ability to escape from local optima. Therefore, CONOP.EA can dig more deeply into previous results to find an even better sequence, which gives it a capability that is not available in the earlier programs.

## Comparison between CONOP.EA and CONOP.SAGA

The performances of CONOP.EA and CONOP.SAGA were compared by using the present dataset with the same parameters: starting temperature = 500, steps = 700, trials = 40,000. Temperature used herein is not an actual temperature or a mean deep-sea temperature, but a parameter in the simulation annealing algorithm[63]. It is notable that the number of trials is much fewer (and less expensive) than those used in the final calculation (i.e., 2,400,000 or 4,800,000) to save time. The result shows that the average penalty of sequences constructed by CONOP.EA is ~538.5 less than that by CONOP.SAGA (Fig. S1a). In the meantime, the average wall-clock time costs of CONOP.SAGA and CONOP.EA are 13 h:09 m:30 s and 33 m:56.56 s, respectively, on an Intel Core i9-10900F processor with 10 cores and a base clock speed of 2.80 GHz, equipped with 32 GB of DDR4 RAM. The performance result demonstrates that CONOP.EA ran ~23 times faster than CONOP.SAGA (estimated over 40,000 trials) by using an updated function (newpen), which calculates only the penalty changes of altered part(s) between the parent and offspring sequences, instead of the penalty of the whole offspring sequence, during each trial, thus highly speeding up the CONOP calculation because this is the most time-consuming aspect. Therefore, CONOP.EA has a better ability to acquire a globally optimised sequence (i.e., lower penalty) and runs much faster than CONOP.SAGA.

Another test is also applied to prove the higher efficiency of CONOP.EA. CONOP.SAGA is first run with the following parameters: starting temperature = 500, steps = 700, trials = 2,400,000 or 4,800,000 to produce ten optimised sequences. Because CONOP.EA has an ability to evolve from previous results; we applied it to the ten

sequences by CONOP.SAGA to generate more optimised offspring sequences with the calculation parameters: starting temperature = 0, steps = 500–700, trials = 240,000. The starting temperature was set at zero so that in each iteration the program will only accept a new sequence with lower penalty, while with a non-zero temperature, the program could randomly (depending on a function including a non-zero temperature) accept worse sequences to escape from local optima. Ten optimised results of the offspring sequences were therefore used to compare with the ten parent sequences. The result shows that the evolved offspring sequences are better than their parent sequences with on average 104.84 fewer penalties (Fig. S1b), proving that this function–evolving from previous results–gives CONOP.EA a unique ability to dig further to find a more globally optimised sequence by recombining various optimal sequences from previous computations. This capability was originally intended for CON-OP.SAGA, but was not successfully realised at the time.

### Analytical procedure of CONOP.EA

The CONOP.EA analysis was implemented in four sequential steps. First, a network analysis of the stratigraphic sections was conducted to determine if all could be connected through co-occurring fossils (Fig. S5). Second, a subset of planktonic foraminiferal index fossils was selected[64,65] and used to test their reliability as time markers under the assumption that these datums were relatively isochronous throughout the world ocean (Fig. S6). The resulting sequence (Table S2) was therefore used to construct a virtual section, which was highly weighted in later computations to speed up the calculation. Third, we ran the parallel computing version of CONOP, CONOP.SAGA ten times. The ten composite sequences present similar trajectories of species richness (Fig. S7 and Table S3) and were then used as parent sequences for CONOP.EA for further calculation and improvement. The optimal (i.e., the one with the lowest penalty) of those generated by CONOP.EA was used in subsequent analyses. Lastly, owing to the lack of sufficient high-precision radiogenic isotope dates for horizons in our dataset, the final composite datum sequence was calibrated to the GTS 2020 time scale through planktonic foraminiferal markers and magnetochrons by a cross-validated smoothing spline (Fig. S8; Table S4; and See Supplementary Note 5).

### Richness estimate and its links with environmental proxy data

An unbinned method was used to calculate the species and genus richness values to avoid losses in temporal resolution from averaging and issues associated with different counting strategies[37] (See Supplementary Note 6). The rarefaction method was utilised to evaluate the effect of sampling on richness estimates (See Supplementary Note 6 and Fig. S9). A binned approach (bin = 0.2 Myr) was also used to calculate proportional origination, extinction, diversification and turnover rates (See Supplementary Note 7). The 95% confidence intervals for richness change rates are derived by iteratively computing change rates following 95% bootstrapped confidence interval of the age model. To assess the association between richness and published, high-resolution environmental proxy data (See Supplementary Notes 8, 9), e.g., carbon and oxygen isotopes, temperature (sea-surface and deep-sea) and sea level[2,11,66,67], Spearman's rank correlation is applied. However, the $\delta^{18}O_{benthic}$ is not used for correlation, as its variance inflation factor (VIF) is larger than 10, indicating potential multicollinearity with other proxies, such as sea level and temperature. Correlations among time series that were linearly detrended (by removing the best-fitting line using the least squares method) are also tested, which emphasises interval-to-interval changes rather than longer-term trends[52].

### Reporting summary

Further information on research design is available in the Nature Portfolio Reporting Summary linked to this article.

## Data availability

All data to reproduce this study are available in Dryad (July 29, 2025 version; https://doi.org/10.5061/dryad.jh9w0vtk5)[68] and Zenodo repositories (Version v2; https://doi.org/10.5281/zenodo.16336191)[69]. The global foraminiferal data were derived from the OneStratigraphy Database (http://onestratigraphy.ddeworld.org), an open-access database. The deposited datasets are publicly available with no access restrictions. Source data are provided with this paper.

## Code availability

All code to reproduce the analyses is provided in the Dryad (July 29, 2025 version; https://doi.org/10.5061/dryad.jh9w0vtk5)[68] and Zenodo repositories (Version v2; https://doi.org/10.5281/zenodo.16336191)[69]. An additional open-access point is provided via the OneStratigraphy Database (https://onestratigraphy.ddeworld.org/download/0f53e05eed3243f1898dc5afd2c54134).

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

## Acknowledgements

We thank Qinghai Zhang for his contribution to verifying the taxonomic assignments in the dataset. We thank Xuan Li, Yuchen Cui, Xiaohong Zhou, Wei Wu, Xiaofang Teng, Yuan Wang and Qin Chen of the ONEs team at Nanjing University for their assistance in data compiling and Jiao Yang for her help in programming and data preparation. We thank Norman MacLeod for numerous helpful discussions during the research and a detailed review on an early draft, and Erin Saupe for constructive comments that helped improve the manuscript. We thank Rongxi Tan for his contribution in refining the CONOP.EA program. We thank the numerous research groups whose published work underpins the stratigraphic sections and fossil occurrences used in this study. The full list of source references and section identifiers is provided in the Supplementary Information and Supplementary Data 2. This research is supported by National Science and Technology Major Project (2022ZD0116600, CQ), National Natural Science Foundation of China grant (92255301, JXF, 42302001, YYD and 624B2069, KX), Natural Environment Research Council grant (NE/V018361/1, BSW), Spanish Ministry of Science, Innovation and Universities grant PID2023-149894OB-I00 (LA), funded by MICIU/AEI/ 10.13039/501100011033 and by the European Union, and Aragon Government grant E33_23R (LA), Nanjing University (14380230, JXF and 14380020, CQ), Jiangsu Science Foundation Leading-edge Technology Program (BK20232003, ZHZ) and International Science and Technology Cooperation Program of Jiangsu (BZ2023068, SZS). This paper is a contribution to the "Deep-time Digital Earth" Big Science Program and OneStratigraphy Database.

## Author contributions

J.X.F., Z.B.L., and Y.K.S. conceived and designed the research. K.X., Y.C.W., X.D.H., Z.B.L., C.Q., Z.H.Z. and J.X.F. designed the CONOP.EA program. Z.B.L., J.X.F., Y.Y.D. and K.X. performed most of the analyses. Z.B.L. and J.X.F. wrote the main text and supplementary information with significant inputs from B.S.W., L.A., M.J.B., C.Q., Y.K.S., P.M.S., Z.H.Z., and S.Z.S. and data contributions from P.Y.F., B.S.W., L.A., and H.Q.X.

## Competing interests

The authors declare no competing interests.
