## [Transparent Peer Review file · Nature Communications]

Complex marine ecological response during the Eocene-Oligocene revealed by global foraminiferal record

Corresponding Author: Professor Junxuan Fan

Version 0:

Reviewer comments:

Reviewer #1

(Remarks to the Author)

Andrew Fraass, University of Victoria

This manuscript describes a new collection of foraminifer occurrences over nearly 30 myrs (sourced from OneStratigraphy), a novel AI algorithm (CONOP.EA), and set of analyses examining the foraminiferal record. This would be, to my memory, the first time that all three groups of foraminifera (larger and smaller benthic (LBF, SBF) and planktic foraminifera (PF)) have been examined together at once, and thus is a novel work on multiple fronts. As it's pretty relevant, I'm specifically a planktic foram specialist that has worked quite often in examining diversity trends in that group, so my understanding of the literature there is quite good, but my knowledge of the LBF, SBF literature is much weaker.

I have a major reservation about this paper. This primary novelty of this paper relies on something the authors have multiple times referred to as novel (CONOP.EA), have very dramatic claims about (a ~26,000-yr resolution on, well, almost anything in the Eocene or Oligocene is extraordinarily impressive), but the paper is very clearly not written about that. CONOP or CONOP.EA only ever appears in the methods or supplemental information. It feels like I'm reading a paper that claims to be a huge leap forward in marine chronostratigraphy, except it's all about the EO boundary diversity trends. Reading the paper I was struck by the saying: "Putting the cart before the horse". The authors have jumped past discussing the innovative work, the algorithm and stratigraphy, or even describing the underlying data in good detail, and leapt right to application of their method. I understand there's a massive incentive to go for a big splashy paper first and then go back and do the less 'exciting' paper second, but there's a rough draft of the CONOP.EA paper sitting as the supplemental for this paper. In fact, almost all of the following is a review of the supplemental, because I have some major reservations about that portion of the work, and without those being resolved, it's difficult to really review the output at the higher level (the paper as written). I apologize if this is overly harsh, but essentially, the Supporting Information is the paper that should be written and reviewed first, and *then* the discussion of foraminiferal diversity trends should happen next. That supporting information paper, however, is lacking in several respects.

Regarding the base data: Looking at the data available in OneStratigraphy, I did a search on "Foraminifera". 55 records appeared. and the dataset appears to be ~1/4th comprised of Triton (Fenton & Woodhouse et al. 2021) and a smattering of continental drill holes. I would like to point out that Fenton & Woodhouse et al. (2021) is not the correct reference for any of these data. That was an aggregation of previously published data, and to not link to the original publication is doing a disservice to the original authors - and authors who come later and want to investigate the original context of those data. This violates FAIR data principles (e.g., <https://www.go-fair.org/fair-principles/f2-data-described-rich-metadata/>) and I would *highly* suggest the folks running the database examine those as they continue on with their project. It looks like that problem is - possibly - only within the presentation of data via the website, but it is not clear how deep this issue persists within the database.

I used the provided link within the supplemental (lines 95-96). The datafile appears to be the bioevents, rather than the raw occurrence data. The raw occurrence data is what these inferences is based on, and while the authors claim it's available on OneStratigraphy, unless I'm missing some organizational difference, there's ~100 localities missing. On the readable datafile, some of the headers are impossible to interpret (e.g., "M?F (M¹refte) A vin 111_base"). While some of this could maybe be due to my excel trying to interpret Chinese characters (potentially?), a key to which specific studies these data were originally sourced from is vital. This could be as simple as a spreadsheet with the header's code and a full citation for

the original publication. As somebody who works with these sorts of data, there's a huge amount of decision making that goes on between the publication and turning it into a single bioevent (e.g., did the original authors use only a *senso stricto* form for their top, while the folks who input the data incorporate *senso lato* forms as a part of their range? Was any marked '?' included). Secondly, as somebody who has generated these kinds of data as well, it *sucks* to be included in these kinds of meta-analyses and not acknowledged at all. This is a broader concern as the entire paleo community does more and more of these projects, but obfuscating (accidentally, I'm sure!) the original publications is a big step backwards.

I am now further in the supplemental, but it actually appears that the raw data being worked with here is not occurrences, but abundance data. While that's a semantic issue, one can't be sure of the sample sizes with just presence/absence data (Supplemental line 364). I'm generally one to make the argument that planktic foraminifers don't require deep considerations of sampling constraints as our 'count to 300' rules of thumb and massive numbers of individuals w/ fairly low numbers of taxa in PF studies, but SBF/LBFs are likely fairly different. In fact, Lloyd et al 2012 in *Paleobiology* would argue that PF are strongly impacted by sampling biases. I appreciated the discussion of rarefaction by the authors. It does go towards answering the questions below about the appropriate numbers of LBF and SBF taxa, but I'm not fully convinced.

There's very little description of the initial data that went into the CONOP.EA program. How many of these localities are from DSDP/ODP/IODP? How many from continental drilling? Outcrops? How many of them are from coastal settings vs. deep sea? The environmental tolerances for many of these organisms aren't overlapping; I certainly don't expect LBFs everywhere I expect PFs for example. How many of the listed taxa are PF, how many LBF, how many SBF? Is this enough to really represent the entire richness of SBFs or LBFs (not worried about PFs, there aren't that many taxa relative to the other two)? The scales on both figure S9 and S11 make it difficult to determine how many PFs, for example, there actually are at peak. I'm relatively convinced that there's enough PFs to represent their record given their figures and discussion, but the huge discrepancy between the Whidden and Jones (peak in the Lutetian is ~2x what W&J report) and lack of any SBF paper to compare to makes me much less certain that the input is of a representative sample. There's very little discussion of the taxonomy within this paper other than for the planktics. SBF and LBFs must have come from WoRMS or the "related taxonomic references", but there's not enough information here to understand what names came from where. As SBF and LBF taxonomy is (largely) less worked out than the (largely) community-standardized PF taxonomy, and given the results within S11, for example, SBF and LBF records strike me as the weakest link here. The next line after the sentence about all of the cleaning and taxonomic work, it says that "These opinions were finally concluded [it is unclear what this means] and verified by a foraminiferal expert (Fang) for correctness and consistency." I will admit, I'm somewhat skeptical that these taxonomic opinions were all well verified if it was just a single member of the author team working on all the PF, SBF, and LBF taxonomy. Those are functionally different specialities, even SBF and PF, despite having lots of samples overlapping, are an uncommon speciality to possess by a single person. I mean this with all due respect to Dr. Fang, but I certainly wouldn't trust my taxonomic opinions - or even my interpretation of a misspelled taxonomic name - when it came to smaller benthic foraminifera.

It's also been clear for a while that while we assume that many of our PF index fossils we assume are isochronous, they're actually not and there's a geographic pattern to their first appearance (e.g., Lam et al, 2022). There's not much, if any, discussion of diachroneity. I'm not of the opinion that the authors were wrong to use the GTS 2020 / Wade et al 2011 calibrations, they should, but the assumption of isochrony, especially with a stated 0.026 myr precision, is unreasonable.

While there is a map in Fig S1, there are not symbols designating which groups of studies are done at each location (SBF, LBF, or PF, or a combination), which where would help, nor the kind of environments, especially as the map is so zoomed out. It appears that there are sites found on the continent of Antarctica, which made me initially think this was a modern map, but upon closer examination it appears to be a reconstruction. There is no information, other than a reference to a Scotese publication, about the map. Turning to the network diagram (Fig S4) doesn't help, there's no clarification on what communities are which colour. Is blue-ish LBFs, while red planktics and the green and brown different regions of benthic foraminifers? The authors claim that there's a relationship to paleogeography and stratigraphy, but this is not further discussed. There are almost certainly a large proportion of these localities where only one group is represented. Was there any control on publications where, for example, PF specialists note a few key benthic foraminifer taxa but aren't considering most of the SBFs? Discussions about how these sites were chosen, what considerations on the broader data compilation was, what proportion of sites were just a single group or multiple, and so forth, would provide a much better picture of the underlying data.

In the supplemental lines 84-88, there's a discussion of Hole 647A, describing how it's been looked at for 'different types of foraminifera for variable use'. This implies that all groups, PF, SBF *and* LBF were found there, but I believe these studies are all on PF and SBF. Rare to find appreciable quantities of LBFs in deep sea drilling material. This paragraph then states that this work was condensed to a single section by depth, but then the last sentence describes that there were 46k occurrences, 9k bioevents from ~1,300 species, in 161 sections. I don't see how the discussion of 647A fits in, unless it's supposed to be an example of integrating different post-cruise work together onto a single section.

It looks like the authors also used GTS 2020 as their numerical calibration; as they've pointed out that there aren't that many horizons dated radiometrically in the dataset (to be expected). This would be a tremendous amount of additional work - probably amounting to a different study - but I suspect that including paleomagnetic stratigraphy found at each site where possible would be very useful. I suspect this would be an entirely different paper, but an effective test of the CONOP.EA results, as that would be an independent set of age diagnostic criteria, and would allow the authors to see which taxa are difficult to resolve (potentially from diachroneity?) or easy to resolve (hopefully the index taxa). I don't want to imply that the logic here is circular, because the key aspect of this project appears to be the order rather than the absolute age (though, absolute ages are important at some level here), the independence and a test of their results is however what this would

solve.

The brief description of the data within the Methods and on Dryad is different, with different amounts of localities and species. This is explained in the supplemental as a part of the cleaning process, but it's jarring when one looks at the original supplemental readme files.

In general the text is well written. In a few places the phrasing is awkward (e.g., 208-209), comments that don't really fit within the scientific literature (e.g., "This is what we planned to do in CONOP.SAGA but failed before."), or shifting point of view (largely written from an impersonal standpoint, but then switches to 'we' midway through the supplemental). There are figures which appear to be made just because they are easy (Fig. S8). That's a pretty neat figure, and should probably be dug into!

There is a lot to like about this paper, I think that the underlying algorithm appears to be a substantial step forward in terms of chronologies here, and given the substantially lower computational cost, has the potential to reach a lot of different users with the output. Producing a huge table of the FAD/LADs of *all* planktic foraminifer taxa would be a boon to the field, just by itself. The figures in the main paper here are well designed, and largely fairly clear (I would really like to see many more tick marks on some of the axes).

Main paper line-by-line

34-35: Saying that this transition is poorly constrained is a bit of an overstatement. It's been worked on for years, including by some of the authors. This paper is described as a major improvement, sure, but it's not "poorly constrained".

41-42: "spectacularly" is unscientific language

44: small [benthic] foraminifera ?

82: foraminifers aren't animals, so 'faunal' is incorrect terminology.

86-88: This statement is either misleading to untrue. Some work is done at combining biozones or using 3 and 6 myr bins (Lloyd et al 2012 in Syst. Biol. or Lloyd et al 2012 in Paleobiology), 1 myr bins (Cárdenas-Rozo & Harries 2016, Lowery et al. 2020), or PF biozones (Fraass et al 2015). Some studies, like Ezard et al 2011, don't bin at all and use a continuous time approach. PF biozones also range much more than just 1-1.45 myr. P0 is 0.04 myr in duration and Palpha is only 0.28 myrs. This also implies that having an uneven binning scheme is less valid. While technically not untrue, there are two studies (I apologize, since I'm on both of these) which have looked at PF diversity alongside other groups - though yes, not other groups of foraminifera (Lowery et al., 2020; Jamson et al., 2022). I'm not asking you to cite all of these papers, I understand there are limitations here, but some of these are worth including (esp. Ezard et al. 2011, which would seem to be a very useful paper to bring in conversation here).

110: Units on the origination rates? I assume it's something along $\text{lineage}^{-1} \text{Ma}^{-1}$

113: Why is the origination rate referred to as proportional here and not elsewhere?

115: the interval here is the Priabonian or just the early Priabonian?

150-151: In particular, connecting up to comments in my main statements, a conversation about the different tendency for provincialism and cosmopolitanism within these different groups would be very useful - and worth digging into.

168-171: To not include Ezard et al. 2011 here is a problem, as it digs directly into this question - with some similar methods I might add (correlations between diversity records and environmental proxies, etc). Further, the second sentence here seems to ask for a single driver, which isn't that reasonable, from my perspective. There's a *huge* literature on this, and while yes, the driving mechanism behind all foram diversity changes hasn't been "fully recognized", we certainly have a number of really good ideas about parts of it.

187-190, 234-236: This isn't that novel a result, as many decades of previous research show that sea-surface temp (again, eg. Ezard et al. 2011), demonstrate this. We're also not looking at independent measures, or that the paper really interrogates the correlation between $\delta^{18}\text{O}_{\text{benthic}}$ and sea-level, which should be - and are - very, very correlated.

237-239: The phrasing here is awkward.

304-305: I understand the authors intent here, but this statement is overly grandiose. One could say that studying the pollen found in annual varves in a lake is a much finer resolution than found here, as just one example.

Supplemental

L189: I think this "dig more deeply" ability of CONOP.EA is the recombination, but it is unclear as written.

(Remarks on code availability)

I reviewed the dataset, but not the code.

Reviewer #2

(Remarks to the Author)

Lu et al present an interesting, well written manuscript exploring the relationship between foraminiferal richness and various abiotic parameters using a novel AI algorithm. This manuscript is unique in that it combines both planktonic and benthic foraminifera something rarely done in the community. Using the newly compiled dataset and AI algorithm the authors show that foraminifera show different responses to the Eocene-Oligocene climatic transition depending on their main ecology with small benthic foraminifera showing opposing trends to the other groups in the late Eocene which the authors hypothesize to be a result of increased export of organic matter to the deep.

This manuscript is well written and provides a novel approach to a longstanding problem, but in its current form does not emphasize enough why this manuscript is novel and unique. The authors spend a lot of time explaining that the species curves generated show remarkable resemblance to those already produced, despite the AI algorithm producing a higher-resolution dataset, I would therefore encourage the authors to take some time and explain why this dataset and results are unique and what the differences between this and published datasets reveal. I believe reframing the manuscript and

addressing the minor points below will improve the manuscript for publication in this journal.

Moderate changes:

Line 73 – the term ecogroups here is slightly confusing in the planktonic foraminifera community ecogroups generally correspond to stable isotope defined depth habitats (see Aze et al, 2011) whilst here the authors refer to much larger delineations. I suggest the authors reword this section to avoid confusion. In addition, I would recommend bringing some information regarding the 3 groups from the Supplementary Material into this section to explain habitat affinities of LBF and SBF and why they are separated as currently this separation seems arbitrary and based on size alone.

Line 86-68 – “Further, complications have only involved one group of foraminifera” – expand on this point as to why and what group of foraminifera the authors are referring, I assume planktonic from the references, why are one group focused on more than others?

Figures 1- 2 – A lot of the text refers to very small intervals throughout, some within the individual stages, these are hard to pinpoint due to the lack of minor tick marks or other visual aid. Consider adding something to both plots. An effort to do this has been done in Figure 1, delineating event numbers but these are then not referred to in the text.

Minor Changes:

Line 57 – p in pCO₂ should be italicized

Line 103 – 06 – this sentence is difficult to read, consider rewording

Line 117 – Include a figure reference here

Line 274 – Is LIP defined before this point? If not then define it here.

Methodology

I think the manuscript would benefit from more information regarding the dataset used such as:
Were the occurrences taken from publications or ODP/IODP reports and how were these chosen?
What is the resolution differences between the groups?
What is the geographical spread of the data?

(Remarks on code availability)

Reviewer #3

(Remarks to the Author)

The manuscript describes a global trajectory of foraminiferal species during the Eocene–Oligocene transition. To do so, it relies on a rich foraminifera fossil record (~45 k fossil occurrences, ~9 k first and last occurrence, ~1.2 k species). The manuscript also proposes a new metaheuristics optimization algorithm (constrained optimization based on evolutionary algorithm, CONOP.EA), improving on CONOP.SAGA. The manuscript is generally well organized and well written, with appropriate figures and tables. I have only minor suggestions.

My only main comment is that I suggest being a bit more specific in the abstract and conclusions that the proposed methodology is based on a metaheuristic optimization algorithm, or an optimization tool that uses AI-inspired metaheuristics. A “novel AI algorithm” is slightly vague, especially now that the term AI is most frequently associated with machine (or deep) learning.

The overall methodology seems sound, however I had trouble using the provided executables as described below.

Minor comments:

Supporting information

Lines 199-200: “In the meanwhile, the average time costs of CONOP.SAGA and CONOP.EA are 40944.83s and 2387.71s, respectively” – I suggest converting these numbers to a more human readable format (hours, minutes, seconds). I also suggest providing information on what CPU was used. Similar for table S2.

Lines 286-287: “We first ran CONOP.SAGA on the ‘Tianhe II’ supercomputer” – I suggest adding more information about the resources used (number of cores used, CPU speed, memory required if easy to estimate, etc).

File EOT_forams_Readable.csv (<https://doi.org/10.5061/dryad.jh9w0vtk5>)

It might be my computer, but question marks appear in some column names when I open the file using text editors (e.g., VS Code, or notepad++, two different computers, Sark?y vin 13_base,Sark?ySark?y vin 13_top,Ke?ili vin 15_base,Ke?ili vin 15_top) so I wonder if there was some encoding conflict when uploading the files or if that was intended?

Executable CONOP++.exe

Unfortunately, I was not able to run the program. I use another SO for work and I only have access to an old Windows laptop, so I don't know if that's what's causing the issue. However, the program did not seem to take my input in the “Enter annealing” input after selecting “Project > Run - TXT”. The program window goes black I select “Project > Run - CHT”.

(Remarks on code availability)

I am not sure if the code itself is available, it seems only the executable is. There is a README file, but I could not use the executable as described in the main review.

Version 1:

Reviewer comments:

Reviewer #1

(Remarks to the Author)

First, let me apologize for the exceedingly slow time for this second review. I got covid twice in 2 months, the first of which was the literal first day of the term. Thus, I've spent the last several months catching up with work.

I was really happy to see Lu et al. 2025 published! Felt like a needed piece of this study's puzzle.

Once the criticism discussed below relating to the Triton data is dealt with, I am fine with this work being published, but that issue must be addressed first.

The authors have done a commendable job dealing with my criticisms, and while I have a few minor things that I would quibble with, some of those are outside the scope of this work. As an example, with respect to syn/diachrony: I understand the rebuttal, but I still fundamentally question the ability to claim having a temporal resolution so fine - my read on this as an (occasional, these days) biostratigrapher is that this claims we now know that FAD G. species A occurs at x Ma +/- 0.029 Myr (0.0145 Myr?). I would suspect in some cases it's possible the dispersal of new planktic species could not ever be resolved as finely. But, again, I do not think this is a criticism that should hold this paper back at all, it's more of a fundamental 'can we actually state confidence levels that small like this given this is a biological system aliased by a slow geologic record' than anything to do with this paper fundamentally.

Thank you for the guidelines on how to retrieve the deeper occurrence information, this is substantially different than what I was looking at before. Having poked through a few of the different entries, yes, the original publication is there - not sure how I must have ended up wherever I was initially in the database. Thanks!

The key criticisms remaining which must be addressed, however, is: What is the number of entries which came originally from the Triton or NSB import? The Triton import would also bring along the NSB and ForCens data with it, as an incomplete set of those data are contained within Triton. When I use the Triton reference in OneStratigraphy to find all the included Triton data, it pulls up 31,539 pages of occurrences, though it's unclear if those were included in this work, as the first page appears to be missing key information for it to be usable (locality is unknown). With a full "foraminifera" search providing 56,495 pages, a rough estimate of the Triton component of this dataset could be on the order of ~55%.

It's possible that I'm looking at this incorrectly and there's certainly a potential for some miscommunication on my side here. Is all the Triton data within OneStratigraphy simply "staged" at the moment and not included in this analysis? Is some of it staged and some included, and if so, how much? The text in the rebuttal re: Triton "However these data will not be used in any formal studies until the metadata are fully completed and verified. The OneStratigraphy team is currently trying to enrich and correct the metadata, including linking each Triton record back to its original publication." From the original Triton publication: "The source of the data (**source**) is recorded based on the data citation and **year** in which it was collected. The primary data sources (e.g. Neptune, Pangaea) are given unique IDs (**db.source**). Individual datasets within this are given unique IDs (**db.ID**); these are particularly relevant for Pangaea where multiple, separate datasets exist. Each site is given a unique **holeID**, and samples within sites are designated using the **sampleID** (which is a unique number added to the **holeID**). The **rowID** is created by combining the database ID, the **sampleID**, and a unique number assigned to each row (i.e. species). The **person** who entered the data and **date** of the most recent update of that entry is also recorded."

Required metadata (more, in fact) appears to be within the original Triton data, unless the authors are looking for something not included here? This leads me to believe that Triton is included, at least partially.

The 55% is obviously a rough and again, I have no doubt, incorrect number, but even some data incorporated here from NSB or Triton is present, the work being built upon must be discussed, acknowledged, and further outlined how the OneStratigraphy data is unique within the text. Especially given this study's comparison to the Fenton and Woodhouse richness curve. If the majority of the underlying data were from Triton, it's no surprise the richness curves were similar. As far as I can tell, there is no discussion of Triton or Neptune by name in the paper, which then relegates them to simply citations for criticisms of them as previous work (e.g., L99-106) and comparison points, rather than as major sources of data, or the major source of data if the 55% is correct. I'd also like to point out that the Fenton et al. paper is a joint first author paper, and so should be cited as Fenton & Woodhouse et al. Further, one of the major sources of information in the Triton dataset is ForCenS, which does not appear in the publication at all.

R1.2 of FAIR is "Metadata are associated with detailed provenance." I understand the difficulty of producing a way of presenting data which has a "NSB->Triton->OneStrat" or "ForCenS->Triton->OneStrat" provenance, or which literature source NSB/Triton pulled data from. A discussion of where this data came from as well as the amount of the data from those sources is important, and any modification of those individual sources.

Without a full accounting of these concerns, it is not possible to recommend this manuscript for publication. If there is no Triton, NSB, or ForCenS data, or any other data compilation incorporated than I recommend this manuscript for publication.

(Remarks on code availability)

Reviewer #3

(Remarks to the Author)

I would like to thank the authors for addressing my comments, I have no further suggestions.

(Remarks on code availability)

The executable CONOP_EA_from_existing.exe worked on my computer (Windows 10, Intel Core i5-7300HQ CPU @ 2.5Ghz, 8 Gb RAM)

Version 2:

Reviewer comments:

Reviewer #1

(Remarks to the Author)

I greatly appreciate the time and efforts of the authors, thank them for the clarifications, and have no further comments. I look forward to seeing this work published.

(Remarks on code availability)

REVIEWER COMMENTS

Reviewer #1 (Remarks to the Author):

Andrew Fraass, University of Victoria

This manuscript describes a new collection of foraminifer occurrences over nearly 30 myrs (sourced from OneStratigraphy), a novel AI algorithm (CONOPEA), and set of analyses examining the foraminiferal record. This would be, to my memory, the first time that all three groups of foraminifera (larger and smaller benthic (LBF, SBF) and planktic foraminifera (PF)) have been examined together at once, and thus is a novel work on multiple fronts. As it's pretty relevant, I'm specifically a planktic forum specialist that has worked quite often in examining diversity trends in that group, so my understanding of the literature there is quite good, but my knowledge of the LBF, SBF literature is much weaker.

1. I have a major reservation about this paper. This primary novelty of this paper relies on something the authors have multiple times referred to as novel (CONOPEA), have very dramatic claims about (a ~26,000-yr resolution on, well, almost anything in the Eocene or Oligocene is extraordinarily impressive), but the paper is very clearly not written about that. CONOP or CONOPEA only ever appears in the methods or supplemental information. It feels like I'm reading a paper that claims to be a huge leap forward in marine chronostratigraphy, except it's all about the EO boundary diversity trends. Reading the paper I was struck by the saying: "Putting the cart before the horse". The authors have jumped past discussing the innovative work, the algorithm and stratigraphy, or even describing the underlying data in good detail, and leapt right to application of their method. I understand there's a massive incentive to go for a big splashy paper first and then go back and do the less 'exciting' paper second, but there's a rough draft of the CONOPEA paper sitting as the supplemental for this paper. In fact,

almost all of the following is a review of the supplemental, because I have some major reservations about that portion of the work, and without those being resolved, it's difficult to really review the output at the higher level (the paper as written). I apologize if this is overly harsh, but essentially, the Supporting Information is the paper that should be written and reviewed first, and *then* the discussion of foraminiferal diversity trends should happen next. That supporting information paper, however, is lacking in several respects.

R: Thank you for raising this important concern. We sincerely apologize for any confusion regarding the focus and structure of our manuscript. This manuscript was first submitted to *Nature*, where strict word count limitations (a maximum of 3000 words) meant that we had to place substantial methodological details, including the development and novelty of CONOP.EA, in the Supplementary Information. At the suggestion of the *Nature* editorial team, we subsequently submitted the manuscript to *Nature Communications* via their internal transfer process. Since *Nature Communications* has more flexible formatting and length guidelines, we have thoroughly revised the manuscript in response to the reviewers' suggestions.

We have added a new section to the main text entitled "Reconstructing high-resolution chronology using CONOP.EA: an AI-inspired metaheuristics algorithm". In this section, we describe the advantages of our newly compiled, global foraminifera dataset and its broader geographic and taxonomic scope. We also explain how combining this dataset with the CONOP.EA algorithm enables the construction of a high-resolution chronological framework for the Eocene–Oligocene transition (EOT). This new section highlights the methodological innovations introduced to improve temporal resolution and data integration. Further technical details are provided in the Methods section for readers interested in examining the implementation in depth.

2. Regarding the base data: Looking at the data available in OneStratigraphy, I did a search on "Foraminifera". 55 records appeared. and the dataset appears to be ~1/4th

comprised of Triton (Fenton & Woodhouse et al. 2021) and a smattering of continental drill holes. I would like to point out that Fenton & Woodhouse et al. (2021) is not the correct reference for any of these data. That was an aggregation of previously published data, and to not link to the original publication is doing a disservice to the original authors - and authors who come later and want to investigate the original context of those data. This violates FAIR data principles (e.g., <https://www.go-fair.org/fair-principles/f2-data-described-rich-metadata/>) and I would **highly** suggest the folks running the database examine those as they continue on with their project. It looks like that problem is - possibly - only within the presentation of data via the website, but it is not clear how deep this issue persists within the database.

R: Thank you for pointing these out.

- Regarding the foraminifera records in OneStratigraphy:

To obtain a complete view of the foraminiferal records in OneStratigraphy, we recommend the following steps: Navigate to “Stratigraphic Data” → “Advanced Search” → “Occurrences” and set the fossil group to “Foraminifera” (see Fig. 1 below). As of 12 June 2025, the OneStratigraphy database contained 826,688 foraminiferal occurrences (see Fig. 2 below), substantially exceeding the size of the Triton dataset.

To avoid this confusion, we have added a detailed description in Supplementary Lines 69-82:

“To obtain a complete view of the foraminiferal records in OneStratigraphy, users are recommended to follow these steps: Navigate to “Stratigraphic Data ” → “Advanced Search ” → “Occurrences ” and set the fossil group to “Foraminifera ” . As of 12 June 2025, the OneStratigraphy database contained 826,688 foraminiferal occurrences. Data used in this study were manually collected from peer-reviewed publications. Each data record includes clear metadata linking back to the original source in the OneStratigraphy database. Any errors were corrected (e.g., spelling mistakes in species

names) and missing information was filled in (e.g., latitude and longitude data). We selected sections/sites containing foraminifera occurrences from the Eocene to the Oligocene. The raw dataset contained 13,138 local bioevents records (i.e., first and last appearance records) and ~60,000 occurrences of 2,988 taxonomic units from 163 published stratigraphic sections, encompassing both calcareous and agglutinated foraminifera.”

- Regarding the Triton database and FAIR principle:

The foraminiferal data used in our study do not originate from the Triton database. Instead, since 2019, we have commissioned a dedicated data entry team to digitize stratigraphic and palaeontological information directly from peer-reviewed publications. Each data record in OneStratigraphy includes clear metadata that links back to the original source. For example, the data from Site 94 (see Fig. 3 below) can be fully traced back to the original publication (i.e., Molina et al., 1993), and this reference is clearly documented in OneStratigraphy (see Fig. 4 below). The same standard is applied to all sections and sites used in our dataset.

The Triton team recognized the great potential of OneStratigraphy for digitizing and hosting global stratigraphic data and decided to collaborate with the OneStratigraphy team to improve the quality of data in the Triton database. The Triton data available in OneStratigraphy is based on this collaboration. However, these data will not be used in any formal studies until the metadata are fully completed and verified. The OneStratigraphy team is currently trying to enrich and correct the metadata, including linking each Triton record back to its original publication. We are committed to ensuring that all data on the platform comply with the FAIR principles before they are used or analyzed formally.

Accordingly, we have added a detailed description regarding this issue in Supplementary Lines 74-82:

“Data used in this study were manually collected from peer-reviewed publications. Each data record includes clear metadata linking back to the original source in the OneStratigraphy database. Any errors were corrected (e.g., spelling mistakes in species names) and missing information was filled in (e.g., latitude and longitude data).”

We appreciate your concern about the FAIR data principles. Ensuring transparency and proper attribution are priorities for us, and we are committed to upholding these standards throughout our work and database development.

Fig. 1. Search workflow in OneStratigraphy

Fig. 2. Retrieved foraminifera occurrences in OneStratigraphy

- Section name: Site94 Molina 1993
- Section reference: Molina et al., 1993
- Tags: (±5) Add Tag
- Is GSSP: No
- Project: Default
- Accessibility: Open
- Section type: Onshore outcrop
- Comments: continental slope of the Yucatan platform

- Country: Mexico
- Province/State: Yucatán
- City/County: (empty)
- Village: (empty)
- Locality: Yucatan platform
- GPS: 24°31'38.4"N, 88°26'9.6"W

Fig. 3. Data overview of Site 94 in OneStratigraphy

- First: Eustoquio Molina
- Second: Concepción Gonzalvo
- Third: Gerta Keller
- Others: First name Last name, First name Last name

- Title: The Eocene-Oligocene planktic foraminiferal transition: extinctions, impacts and hiatuses
- Editor: (empty)
- Book/Report: (empty)
- Journal: Geological Magazine
- Volume: 130
- Page no.: 483, 499
- Publisher: (empty)
- Language: English
- No.: 4
- Year: 1993
- City: (empty)
- Ref. type: journal article

 The 'Other' section includes:

- Comments: (empty)
- Accessibility: Open
- Upload files: Open upload page

 At the bottom, there are buttons for 'Save changes', 'Section included', 'Occurrence included', 'Taxonomy included', and 'Export citation'.

Fig. 4. Traceable reference source for Site 94 in OneStratigraphy

3. I used the provided link within the supplemental (lines 95-96). The datafile appears to be the bioevents, rather than the raw occurrence data. The raw occurrence data is what these inferences is based on, and while the authors claim it's available on OneStratigraphy, unless I'm missing some organizational difference, there's ~100

localities missing. On the readable datafile, some of the headers are impossible to interpret (e.g., "M?F (M"lrefte) A vin 111_base"). While some of this could maybe be due to my excel trying to interpret Chinese characters (potentially?), a key to which specific studies these data were originally sourced from is vital. This could be as simple as a spreadsheet with the header's code and a full citation for the original publication. As somebody who works with these sorts of data, there's a huge amount of decision making that goes on between the publication and turning it into a single bioevent (e.g., did the original authors use only a senso stricto form for their top, while the folks who input the data incorporate senso lato forms as a part of their range? Was any marked '?' included). Secondly, as somebody who has generated these kinds of data as well, it *sucks* to be included in these kinds of meta-analyses and not acknowledged at all. This is a broader concern as the entire paleo community does more and more of these projects, but obfuscating (accidentally, I'm sure!) the original publications is a big step backwards.

R: Thank you very much for this suggestion. We have now added Table S5, which lists all fossil occurrences along with the relevant references. These data can also be accessed via the OneStratigraphy platform by querying the section or site names listed in the table.

We apologize for any confusion caused by the erroneous headers. These have now been corrected in the updated readable data file and also Table S6 with standard characters being used to prevent issues with interpretation.

We fully agree with you about the importance of acknowledging the contributions of original studies (publications). This is why, in the OneStratigraphy platform, we require the original source publication to be clearly cited. This demonstrates respect for the contributions of the original data producers and provides users with critical information with which to assess the quality of the data. To further demonstrate our respect and acknowledgement, we have added the following description in the Supplementary

Information (Lines 74-76) and have retained the source reference information for each data record in multiple tables such as Tables S5 and S6:

“Data used in this study were manually collected from peer-reviewed publications. Each data record includes clear metadata linking back to the original source in the OneStratigraphy database.”

4. I am now further in the supplemental, but it actually appears that the raw data being worked with here is not occurrences, but abundance data. While that's a semantic issue, one can't be sure of the sample sizes with just presence/absence data (Supplemental line 364). I'm generally one to make the argument that planktic foraminifers don't require deep considerations of sampling constraints as our 'count to 300' rules of thumb and massive numbers of individuals w/ fairly low numbers of taxa in PF studies, but SBF/LBFs are likely fairly different. In fact, Lloyd et al 2012 in Paleobiology would argue that PF are strongly impacted by sampling biases. I appreciated the discussion of rarefaction by the authors. It does go towards answering the questions below about the appropriate numbers of LBF and SBF taxa, but I'm not fully convinced.

R: Thank you for your comment. We agree with your concern about the sampling biases. Therefore, in this study, we have adopted two methods to address this issue, including CONOP and rarefaction. Sampling biases can greatly affect our observations on the true stratigraphic level of bioevents, potentially leading to the underestimation of some fossil ranges. This can be corrected using the principles of deterministic biostratigraphy (e.g., CONOP), wherein the ends of incomplete ranges in local sections/sites can be extended to conform to the global event by using observed co-existences and the total stratigraphic range of each species in the entire dataset (i.e., the lowest of all local first appearances and the highest of all local last appearances). This thereby corrects for regional diachronism caused by migration, fossil preservation, and sampling biases (e.g., limited abundance), estimating the most complete species ranges in the dataset (Sadler et al., 2003, 2004; Cody et al., 2008). On this basis, we also applied rarefaction

to test the sampling effect of our dataset by making a standardized comparison of unevenness in species records in the full foraminiferal curve and in the curves of the three foraminiferal groups.

To make this clear, we added the following description in Lines 171-173:

“The Constrained Optimization (CONOP) compositing method⁴¹ is used to integrate the local biostratigraphic data from all 161 sections/sites and to correct regional diachronism caused by migration, fossil preservation, and sampling biases.”

5. There's very little description of the initial data that went into the CONOP.EA program. How many of these localities are from DSDP/ODP/IODP? How many from continental drilling? Outcrops? How many of them are from coastal settings vs. deep sea? The environmental tolerances for many of these organisms aren't overlapping;

R: Thank you for your valuable comments. We have added the following detailed descriptions of the dataset to the section of “Reconstructing high-resolution chronology using CONOP.EA: an AI-inspired metaheuristics algorithm” (Lines 146-170):

“Our assembled global foraminifera dataset (raw dataset) comprises 13,138 local bio-events records (i.e., first and last appearance records) and ~60,000 occurrences of 2,988 taxonomic units from 163 published stratigraphic sections, encompassing both calcareous and agglutinated foraminifera. After the cleaning process (see SI), the final dataset contains ~40,000 fossil occurrences of 1,269 species from 161 drill cores and outcrops (Fig. S1; Table S6). Stratigraphic data of species of three foraminiferal groups, including 277 PF species, 340 LBF species and 652 SBF species, are involved in the dataset. Of the 161 sections/sites, 102 yield one foraminiferal group, 59 yield at least two groups, and 16 include all three groups, covering major ocean basins, including the Atlantic and Indian Oceans, as well as key marginal seas such as the Mediterranean, Caribbean, and Gulf of Mexico (Fig. S1). Occurrence data were extracted from original biostratigraphic distribution records (depth-versus-taxon) reported in peer-reviewed journal articles and scientific reports, including 12 continental drill cores, 48

DSDP/ODP drill cores, five ocean drill cores, and 96 outcrops. In addition, 18 magnetochrons from five GSSP or GSSP candidate sections (Massignano, Oyambre, Agost, Varignano, and Alano) were incorporated into the dataset to enhance stratigraphic correlation and ensure robust age calibration. Within this dataset, 77 outcrops/cores (48%) are assigned to coastal settings, e.g., shallow marine, inner to middle shelf, neritic, and lagoon, reflecting deposition within the photic zone and proximal to land (Table S6). Sixty-nine outcrops/cores (43%) were attributed to deep marine settings, i.e., bathyal, abyssal, continental slope, and pelagic environments, representing deposition below the photic zone and distal from siliciclastic input. The remaining 15 outcrops/cores (9%) exhibited mixed shallow–deep characteristics, indicating transitional depositional settings along the shelf - to - basin gradient.”

6. I certainly don't expect LBFs everywhere I expect PFs for example. How many of the listed taxa are PF, how many LBF, how many SBF? Is this enough to really represent the entire richness of SBFs or LBFs (not worried about PFs, there aren't that many taxa relative to the other two)?

R: Thank you for your comment. Our dataset contains the following taxa for the study interval: 277 planktic foraminifera (PF), 340 larger benthic foraminifera (LBF), and 652 small benthic foraminifera (SBF). We have also added a description in Lines 151-153:

“Stratigraphic data of species of three foraminiferal groups, including 277 PF species, 340 LBF species and 652 SBF species, are involved in the dataset.”

As shown in Fig. S9 (as revised Fig. S7), our LBF diversity curve includes nearly twice as many taxa as that of Whidden and Jones (2012). For SBF, the number of taxa in our dataset also exceeds that of Alegret et al. (2021), who focused on a subset of dominant species for quantitative abundance analysis. In contrast, our study emphasizes diversity patterns and thus incorporates a broader range of taxa. These comparisons suggest that our dataset is currently the most comprehensive and representative collection of both

LBF and SBF taxa.

7. The scales on both figure S9 and S11 make it difficult to determine how many PFs, for example, there actually are at peak. I'm relatively convinced that there's enough PFs to represent their record given their figures and discussion, but the huge discrepancy between the Whidden and Jones (peak in the Lutetian is ~2x what W&J report) and lack of any SBF paper to compare to makes me much less certain that the input is of a representative sample.

R: Thank you for your comment. We have added more scales to Figs. S9 (revised Fig. S7) and S11 (revised Fig. S8) to improve readability. The curve of Whidden and Jones (which includes only three families of larger benthic foraminifera) likely underestimates total richness, explaining why our peak values are nearly twice theirs. Despite the difference in sample size, both curves show a similar overall decline across the Eocene–Oligocene transition, a pattern that remains robust in our subsampled curve (revised Fig. S9).

To date, no comprehensive richness curve for SBF has been published. However, Alegret et al. (2021) presented a Fisher- α diversity curve based on ~14 sections/sites covering our study interval. Their trend closely matches ours—most notably, the Priabonian diversity increase observed in the Pacific Ocean—which lends confidence to our SBF richness record. A corresponding comparison is described in Lines 318-329 of the main manuscript:

“Alegret et al.²² also observed a comparable general pattern of SBF diversity, with a moderate increase in the early Eocene, followed by a decrease during cooling periods, besides a diversity increase in the Pacific Ocean, attesting the establishment of the latitudinal gradient of SBF richness during the early Lutetian. Our results show that a slow richness increase occurred in the early Lutetian, ending with a small drop in richness in the late Lutetian. After a long interval of stable richness values in the

Bartonian, SBF were characterized by a quick (~1.89 Myr) richness increase that more than doubled Bartonian values (i.e., EPR), owing to an interval of extremely high proportional origination rates (more than nine times higher than the average) (Fig. S8a, d). This similar increase can also be observed in the Pacific Ocean²².

8. There's very little discussion of the taxonomy within this paper other than for the planktics. SBF and LBFs must have come from WoRMS or the "related taxonomic references", but there's not enough information here to understand what names came from where. As SBF and LBF taxonomy is (largely) less worked out than the (largely) community-standardized PF taxonomy, and given the results within S11, for example, SBF and LBF records strike me as the weakest link here. The next line after the sentence about all of the cleaning and taxonomic work, it says that "These opinions were finally concluded [it is unclear what this means] and verified by a foraminiferal expert (Fang) for correctness and consistency." I will admit, I'm somewhat skeptical that these taxonomic opinions were all well verified if it was just a single member of the author team working on all the PF, SBF, and LBF taxonomy. Those are functionally different specialities, even SBF and PF, despite having lots of samples overlapping, are an uncommon speciality to possess by a single person. I mean this with all due respect to Dr. Fang, but I certainly wouldn't trust my taxonomic opinions - or even my interpretation of a misspelled taxonomic name - when it came to smaller benthic foraminifera.

R: Thank you very much for raising this issue. In the early stages of this study, we used WoRMS to revise the taxonomic names, and Dr. Peiyue Fang further verified the entire dataset. After that, we did not organize any further efforts to improve the quality of the dataset, as this would have required us to redo all the calculations. However, as you pointed out, this may have resulted in the SBF and LBF data quality being relatively weaker. Therefore, during this revision process, we invited three senior foraminifera specialists to review and validate the taxonomic assignments in the dataset further, including:

Prof. Bridget Wade (PF)

Prof. Laia Alegret (SBF)

Research Prof. Qinghai Zhang (LBF and SBF)

Based on these, we reran all calculations. While the major patterns we previously identified still exist and are significant, they are now derived from a higher-quality dataset, making them more reliable.

Accordingly, we revised the following description in Lines 573-576:

“These opinions were finally verified and resolved by a group of foraminiferal taxonomic experts for correctness and consistency: Bridget Wade (PF), Laia Alegret (SBF), Qinghai Zhang (LBF and SBF), and Peiyue Fang (PF, LBF and SBF).”

9. It's also been clear for a while that while we assume that many of our PF index fossils we assume are isochronous, they're actually not and there's a geographic pattern to their first appearance (e.g., Lam et al, 2022). There's not much, if any, discussion of diachroneity. I'm not of the opinion that the authors were wrong to use the GTS 2020 / Wade et al 2011 calibrations, they should, but the assumption of isochrony, especially with a stated 0.026 myr precision, is unreasonable.

R: Thank you for raising this concern. Yes, we agree with you that even the PF index fossils are not exactly isochronous. This is why we use quantitative stratigraphic methods such as CONOP in this study. In traditional biostratigraphic correlation, researchers typically select a small subset of fossil taxa—those widely recognized as having relatively good isochrony—to define biozones. These index fossils usually comprise only 1–10% of the total fossil species. In contrast, methods like CONOP aim to utilize all available fossil records to estimate the global first and last appearances of each taxon. The algorithm then integrates these appearances across multiple sections to construct an optimized composite sequence. The resulting average resolution appears high because all taxa are retained in the calculations and used for stratigraphic

correlation, rather than just the few that are deemed to be the most isochronous. This broader data inclusion enhances statistical resolution, but does not imply that every individual taxon has perfect isochrony.

Since CONOP can utilize all fossil species, as well as other stratigraphic correlation markers such as geochemical excursions, magnetochrons and sedimentary marker beds, to establish automatic stratigraphic correlation, the time scale it produces is naturally 10 to 100 times more precise than that of traditional biostratigraphy. The resolution of traditional biostratigraphic correlation can be determined by dividing the total duration of the study interval by the number of biozones. Similarly, the temporal resolution of the CONOP-derived time scale can be calculated by dividing the total duration of the study interval by the number of unique first and last occurrences identified. Please note that some first and last occurrences may fall within the same level after the CONOP calculation, meaning this number will be less than the total number of species' first and last occurrences. In this study, the total duration of the study interval is 28 Myr. After CONOP calculations, the total number of unique correlation levels is 962, giving a computed average time resolution of 0.029 Myr. As this temporal resolution differs from that of radioisotopic dating, the term 'imputed resolution' is typically used to distinguish between the two.

In addition to using CONOP to address the issue of fossil diachrony and establish reliable stratigraphic correlations, this study also attempts to quantitatively evaluate the diachrony of commonly used index fossils in order to assess the reliability of prior knowledge in this area. We conducted a non-weighted CONOP analysis on a targeted subset of the dataset consisting only of sections that include the relevant index fossils. The results show that the stratigraphic sequence of these index fossils in the GTS 2020 is fully supported by our dataset. A detailed description of this analysis has been added in Supplementary Lines 283-298 of the revised manuscript:

“The virtual section is composed of planktonic foraminiferal index fossils and

*magnetochrons. The order of their first and last appearance levels, which we refer to as the “virtual sequence”, comes from widely accepted stratigraphic standards such as GTS 2020^{25,26} and Wade et al.²⁷. The order of these fossils was further evaluated by running CONOP on a small subset of the study dataset which only contains sections with these index fossils. Magnetochrons are tested in five GSSP and GSSP candidate sections containing them for the consistency with index fossils. If there are any inconsistencies between the CONOP results/real sections and the virtual sequence, we should refer back to the original stratigraphic report to determine the source of the problem. In our experience, inconsistencies mainly arise from incompleteness of fossil ranges in sections (even in GSSP sections). For example, the last appearance of *Catapsydrax dissimilis* in Hole 1130A&C32 occurs at a lower level than that recognized in the planktonic foraminifera zonation²⁷. Checking the original publication reveals that this site has a low recovery, with the species *C. dissimilis* only found in one sample and being rare. Therefore, the virtual sequence after verification is robust not only in previous studies but also in the present dataset (Table SI).”*

Moreover, according to your insightful suggestion, we have incorporated 18 magnetochrons from five GSSP or GSSP candidate sections to enhance stratigraphic correlation and refine the construction of the age model. These are generally considered to be globally synchronous. Furthermore, we applied a cross-validated smoothing spline to derive the final age model. This method does not force the age model to pass exactly through each tie point, allowing it to accommodate potential diachroneity in index fossil horizons. As such, it reduces the impact of assuming strict isochrony and provides a more robust and flexible age framework. We believe this approach adequately addresses the concern raised and ensures the reliability of our age model, despite the limitations of biostratigraphic precision. The relevant details have been added in Lines 160-162 and Lines 665-668:

“In addition, 18 magnetochrons from five GSSP or GSSP candidate sections (Massignano, Oyambre, Agost, Varignano, and Alano) were incorporated into the

dataset to enhance stratigraphic correlation and ensure robust age calibration.”

“...the final composite datum sequence was calibrated to the GTS 2020 time scale through planktonic foraminiferal markers and magnetochrons by a cross-validated smoothing spline (Fig. S6; Table S3).”

10. While there is a map in Fig S1, there are not symbols designating which groups of studies are done at each location (SBF, LBF, or PF, or a combination), which where would help, nor the kind of environments, especially as the map is so zoomed out. It appears that there are sites found on the continent of Antarctica, which made me initially think this was a modern map, but upon closer examination it appears to be a reconstruction. There is no information, other than a reference to a Scotese publication, about the map. Turning to the network diagram (Fig S4) doesn't help, there's no clarification on what communities are which colour. Is blue-ish LBFs, while red planktics and the green and brown different regions of benthic foraminifers? The authors claim that there's a relationship to paleogeography and stratigraphy, but this is not further discussed. There are almost certainly a large proportion of these localities where only one group is represented. Was there any control on publications where, for example, PF specialists note a few key benthic foraminifer taxa but aren't considering most of the SBFs? Discussions about how these sites were chosen, what considerations on the broader data compilation was, what proportion of sites were just a single group or multiple, and so forth, would provide a much better picture of the underlying data.

R: Thank you for your valuable suggestions. Firstly, in the revised manuscript, we have updated Fig. S1 to include colored circles indicating which foraminiferal groups (SBF, LBF or PF) are represented at each site or section. To avoid any confusion arising from paleogeographic reconstructions, we have chosen to use a modern map as the basis for this figure.

Secondly, we apologize for the lack of clarity in Fig. S4 (now Fig. S3). This figure represents a network diagram based on sites/sections, where the communities are not

grouped by foraminiferal types, but rather reflect paleogeographic provinces sharing similar fossil assemblages. Deng, Fan et al. (2021, ESR) first introduced this figure in a CONOP analysis to demonstrate biostratigraphic connectivity among sections, which serves as the foundation for subsequent CONOP-based stratigraphic correlations. For example, if a few sections have no connections to the others, indicating that they do not share any common fossils, it will be impossible to correlate these sections with the others, no matter how the calculations are performed. These sections will then be removed from the dataset. We generate this figure while preparing the dataset to visualise the internal connections between the section data. Therefore, this figure does not show the distribution of different fossil groups across sections. To examine the distribution of fossil groups at individual sites, please refer to the updated Fig. S1 instead. We have updated the text in Supplementary Lines 260-269 and the caption of Fig. S3 in Supplementary Lines 566-570 to clarify this point:

“The strong connectivity across the dataset supports its suitability for subsequent CONOP-based stratigraphic correlation, as all sections/sites are biostratigraphically linked. Furthermore, the nodes (sections) in the network were grouped into five communities (Modularity = 0.419) by using the “fast unfolding of communities” method²¹ and shown in different colours in Fig. S3. These communities can be interpreted as approximations of biogeographic provinces²². Nodes within each community share similar fossil assemblages, reflecting comparable palaeogeographic and stratigraphic relationships. While this observation opens opportunities for future research into biogeographic connectivity using our high-resolution dataset, such analyses are beyond the scope of the present study.”

“These communities, displayed in different colours in the figure, can be interpreted as approximations of biogeographic provinces²². Nodes within each community share similar fossil assemblages, reflecting comparable paleogeographic and stratigraphic relationships.”

All sections and sites included in this study are derived from peer-reviewed publications.

As you have pointed out, specialists in one group (e.g., PF) may note a few key benthic foraminifer taxa but aren't considering most of the SBFs. We did our best to integrate multiple publications studying the same site or section (e.g., Hole 647 and Massignano), ensuring that all published data on different foraminiferal groups were incorporated into our dataset. As a result, of the sections/sites included in this study, 102 contain only one foraminiferal group, 59 contain at least two groups, and 16 include all three groups.

Accordingly, we have added the following descriptions in Lines 146-160:

“Our assembled global foraminifera dataset (raw dataset) comprises 13,138 local bio-events records (i.e., first and last appearance records) and ~60,000 occurrences of 2,988 taxonomic units from 163 published stratigraphic sections, encompassing both calcareous and agglutinated foraminifera. After the cleaning process (see SI), the final dataset contains ~40,000 fossil occurrences of 1,269 species from 161 drill cores and outcrops (Fig. S1; Table S6). Stratigraphic data of species of three foraminiferal groups, including 277 PF species, 340 LBF species and 652 SBF species, are involved in the dataset. Of the 161 sections/sites, 102 yield one foraminiferal group, 59 yield at least two groups, and 16 include all three groups, covering major ocean basins, including the Atlantic and Indian Oceans, as well as key marginal seas such as the Mediterranean, Caribbean, and Gulf of Mexico (Fig. S1). Occurrence data were extracted from original biostratigraphic distribution records (depth-versus-taxon) reported in peer-reviewed journal articles and scientific reports, including 12 continental drill cores, 48 DSDP/ODP drill cores, five ocean drill cores, and 96 outcrops.”

11. In the supplemental lines 84-88, there's a discussion of Hole 647A, describing how it's been looked at for 'different types of foraminifera for variable use'. This implies that all groups, PF, SBF *and* LBF were found there, but I believe these studies are all on PF and SBF. Rare to find appreciable quantities of LBFs in deep sea drilling material. This paragraph then states that this work was condensed to a single section by depth, but then the last sentence describes that there were 46k occurrences, 9k bioevents from

~1,300 species, in 161 sections. I don't see how the discussion of 647A fits in, unless it's supposed to be an example of integrating different post-cruise work together onto a single section.

R: Sorry for the confusion. You are right that only PF and SBF were analyzed at this site, as detailed in three separate reports. The last sentence you cited from the manuscript was describing the entire dataset, instead of Hole 647A.

To make this clearer, we have revised the text as follows and split it into two paragraphs in Supplementary Lines 108-115:

“In the present dataset, the drill core Hole 647A has been studied repeatedly, focusing on both SBF and PF for variable use, such as testing biotic response to EOT, studying high-latitude deep-water sedimentary sequence, and stratigraphic correlation¹⁷⁻¹⁹. The three reports¹⁷⁻¹⁹ were integrated into one section by depth.

The final dataset after data cleaning and verification included ~40,000 fossil occurrences and 9,032 local first and last occurrence records of 1,269 species in 161 published sections.”

12. It looks like the authors also used GTS 2020 as their numerical calibration; as they've pointed out that there aren't that many horizons dated radiometrically in the dataset (to be expected). This would be a tremendous amount of additional work - probably amounting to a different study - but I suspect that including paleomagnetic stratigraphy found at each site where possible would be very useful. I suspect this would be an entirely different paper, but an effective test of the CONOP.EA results, as that would be an independent set of age diagnostic criteria, and would allow the authors to see which taxa are difficult to resolve (potentially from diachroneity?) or easy to resolve (hopefully the index taxa). I don't want to imply that the logic here is circular, because the key aspect of this project appears to be the order rather than the absolute age (though, absolute ages are important at some level here), the independence and a test of their results is however what this would solve.

R: Thank you for this important suggestion. We have recently completed a technical paper that introduces a new approach to incorporating magnetostratigraphic data into CONOP analyses (Lu et al., 2025. Improving the temporal resolution of middle Eocene–late Oligocene foraminiferal biomagneto-chronology: Insights from CONOP and chronologic significance of biotic events. *Palaeogeography, Palaeoclimatology, Palaeoecology* 669: 112929. <https://doi.org/10.1016/j.palaeo.2025.112929>).

Following your suggestion, we applied this method to the present study, incorporating 18 magnetochrons from five GSSP or GSSP candidate sections to enhance age calibration and improve the overall temporal resolution. This is further described in Line 160-162 and Lines 665-668:

“In addition, 18 magnetochrons from five GSSP or GSSP candidate sections (Massignano, Oyambre, Agost, Varignano, and Alano) were incorporated into the dataset to enhance stratigraphic correlation and ensure robust age calibration.”

“...the final composite datum sequence was calibrated to the GTS 2020 time scale through planktonic foraminiferal markers and magnetochrons by a cross-validated smoothing spline (Fig. S6; Table S3).”

13. The brief description of the data within the Methods and on Dryad is different, with different amounts of localities and species. This is explained in the supplemental as a part of the cleaning process, but it's jarring when one looks at the original supplemental readme files.

R: Sorry for the confusion raised. There are two datasets described in the manuscript: one is the original dataset, and a cleaned, taxonomically standardized version derived from it, which is used for the CONOP calculations. The two instances you pointed out correspond to these two datasets, respectively.

To make this clearer, we have added qualifiers and corresponding explanations in the Methods section. Please see the updated descriptions in Lines 146-151:

“Our assembled global foraminifera dataset (raw dataset) comprises 13,138 local bio-events records (i.e., first and last appearance records) and ~60,000 occurrences of 2,988 taxonomic units from 163 published stratigraphic sections, encompassing both calcareous and agglutinated foraminifera. After the cleaning process (see SI), the final dataset contains ~40,000 fossil occurrences of 1,269 species from 161 drill cores and outcrops (Fig. S1; Table S6).”

14. In general the text is well written. In a few places the phrasing is awkward (e.g., 208-209), comments that don't really fit within the scientific literature (e.g., "This is what we planned to do in CONOP.SAGA but failed before."), or shifting point of view (largely written from an impersonal standpoint, but then switches to 'we' midway through the supplemental). There are figures which appear to be made just because they are easy (Fig. S8). That's a pretty neat figure, and should probably be dug into!

R: Thank you for your helpful suggestions. We have revised the phrasing you mentioned, as well as other instances that we found. The sentence you highlighted has been removed from the manuscript. Throughout the manuscript and supplementary material, we have adopted a consistent, impersonal tone and eliminated the use of "we". Additionally, Fig. S8 has been integrated into the revised Fig. 2a to facilitate direct comparison with the species and genus richness curves. We have also added some discussions of this curve in Lines 291-292 and Lines 293-295:

“Nevertheless, an increase in species/genus ratio and proportional origination rate in the Aquitanian, implies the start of an early Miocene recovery (Fig. 2a, d).”

“The species/genus ratio generally shows a comparable trend to that of the richness change (Fig. 2a), implying that species richness varied while genus richness remained relatively stable. The average value of the ratio is ~1.6 throughout the study interval.”

15. There is a lot to like about this paper, I think that the underlying algorithm appears to be a substantial step forward in terms of chronologies here, and given the substantially lower computational cost, has the potential to reach a lot of different users

with the output. Producing a huge table of the FAD/LADs of *all* planktic foraminifer taxa would be a boon to the field, just by itself. The figures in the main paper here are well designed, and largely fairly clear (I would really like to see many more tick marks on some of the axes).

R: Thank you very much for your thoughtful and encouraging comments. Your insights, especially the professional concerns raised earlier in your review, have significantly helped us refine both the methodology and the presentation of our work. We are truly grateful for your support and your recognition of the potential impact of this approach.

One of our ongoing projects is to incorporate all Cenozoic foraminiferal data into the construction of a new, high-resolution Cenozoic time scale, as well as the evolutionary history of foraminifera based on it. We have been collecting data for over a year, and expect to begin the calculations in the summer of 2026.

As suggested, we have added some tick marks to Figs. 1 and 2 (revised Figs. 2 and 3) to improve visual clarity.

Main paper line-by-line

34-35: Saying that this transition is poorly constrained is a bit of an overstatement. It's been worked on for years, including by some of the authors. This paper is described as a major improvement, sure, but it's not "poorly constrained".

R: Thank you for your comment. We have modified the expression in Lines 35-37 as follows:

“Understanding how the marine biosphere responded during this transition is not well-constrained, appearing as a simple extinction pulse in low temporal resolution global compendia.”

41-42: "spectacularly" is unscientific language

R: Thank you for your comments. We have modified “*spectacularly*” as “*remarkably*”

in Line 44.

44: small [benthic] foraminifera ?

R: Yes. We have written “*small benthic foraminifera*” in Line 47.

82: foraminifers aren't animals, so 'faunal' is incorrect terminology.

R: Thank you for your suggestion. We have deleted this word and ensure that it is not used elsewhere in the manuscript.

86-88: This statement is either misleading to untrue. Some work is done at combining biozones or using 3 and 6 myr bins (Lloyd et al 2012 in Syst. Biol. or Lloyd et al 2012 in Paleobiology), 1 myr bins (Cárdenas-Rozo & Harries 2016, Lowery et al. 2020), or PF biozones (Fraass et al 2015). Some studies, like Ezard et al 2011, don't bin at all and use a continuous time approach. PF biozones also range much more than just 1-1.45 myr. P0 is 0.04 myr in duration and Palpha is only 0.28 myrs. This also implies that having an uneven binning scheme is less valid. While technically not untrue, there are two studies (I apologize, since I'm on both of these) which have looked at PF diversity alongside other groups - though yes, not other groups of foraminifera (Lowery et al., 2020; Jamson et al., 2022). I'm not asking you to cite all of these papers, I understand there are limitations here, but some of these are worth including (esp. Ezard et al. 2011, which would seem to be a very useful paper to bring in conversation here).

R: Thank you very much for your suggestions. Accordingly, we have added the following discussion in Lines 102-111, and deleted the inaccurate description of the temporal resolutions of previous work:

“While previous compilations have mostly focused on planktonic foraminifera³⁰⁻³², including analyses of their ecologically driven diversity dynamics³³ and comparisons with other microfossil groups such as diatoms and calcareous nannofossils^{32,34}, comparative studies involving benthic foraminifera—descendants of the same ancestral lineage but with distinct ecological characteristics—remain limited due to challenges

such as age calibration difficulties, taxonomic inconsistencies, and high levels of endemism. Furthermore, most richness reconstructions rely on binning schemes with uneven or highly variable temporal resolutions^{30,35,36}, though a few studies have applied unbinned methods³³.

110: Units on the origination rates? I assume it's something along $\text{lineage}^{-1} \text{Ma}^{-1}$

R: Yes, and we have added the unit “/Lmyr,” denoting species per lineage per million years, wherever rates are mentioned. The first occurrence of this unit (in Lines 264-266) is now written out in full:

“The decrease was generally gradual with very low proportional origination rate (0.13/Lmyr on average, where /Lmyr denotes species per lineage per million years; Fig. 2d)”

113: Why is the origination rate referred to as proportional here and not elsewhere?

R: Sorry for the confusion. All rates mentioned are proportional, as shown in the figure. To improve clarity, we have added “proportional” to the relevant words in the main text.

115: the interval here is the Priabonian or just the early Priabonian?

R: Sorry for the confusion. This interval is the early Priabonian. We have modified the expression in Line 269-272 as follows:

“During the early Priabonian, species richness doubled, and the proportional origination rate reached two peaks of ~ 0.9 and ~ 0.8 /Lmyr and maintained a consistently high level (Fig. 2d).”

150-151: In particular, connecting up to comments in my main statements, a conversation about the different tendency for provincialism and cosmopolitanism within these different groups would be very useful - and worth digging into.

R: Thank you very much for your insightful suggestion. In this manuscript, we primarily focus on the development of the new dataset and algorithm, the reconstruction

of a high-resolution richness curve, and the analysis of diversity changes in different foraminiferal groups, along with their potential environmental associations. Once this study has been published, we plan to write one or two additional manuscripts. These will cover spatial patterns of those major events during the Eocene-Oligocene transition, as well as investigating contrasting trends of provincialism and cosmopolitanism among the different foraminiferal groups.

168-171: To not include Ezard et al. 2011 here is a problem, as it digs directly into this question - with some similar methods I might add (correlations between diversity records and environmental proxies, etc). Further, the second sentence here seems to ask for a single driver, which isn't that reasonable, from my perspective. There's a *huge* literature on this, and while yes, the driving mechanism behind all foram diversity changes hasn't been "fully recognized", we certainly have a number of really good ideas about parts of it.

R: Thank you for this valuable feedback. Yes, the work by Ezard et al. (2011) is important and insightful, and we have cited it here. We did not express ourselves clearly in the second sentence. Rather than emphasizing a single driving factor, we intended to highlight the combined effects of multiple factors. In other words, different factors affect different oceanic environments and the organisms within them, and this complex mechanism has not been sufficiently elucidated in previous studies. Therefore, we have revised the sentence to make this clearer. Please see Lines 337-341:

“Foraminifera, especially those inhabiting different depths in the water column, appear to show differentiated diversity changes corresponding to related environmental changes^{21,22,31,33,50}. This complex differentiation mechanism involving multiple environmental factors has not been fully elucidated yet in deep-time studies.”

And yes, there are some quite good work investigating the driving mechanism behind foraminifera diversity changes. We have cited those relevant studies in the revised manuscript, for example in Lines 358-361 and 407-410:

“Our results indicate that sea-surface temperature and eustatic sea-level changes correlate with fluctuations in richness for planktonic³³ and larger benthic foraminifera over the entire 28-Myr study interval²⁷.”

“Our results indicate that sudden PF and LBF richness decreases near the Eocene/Oligocene boundary may be related to the combination of sea-surface temperature decrease and eustatic sea-level fall^{27,28} (Fig. 3).”

187-190, 234-236: This isn't that novel a result, as many decades of previous research show that sea-surface temp (again, eg. Ezard et al. 2011), demonstrate this. We're also not looking at independent measures, or that the paper really interrogates the correlation between $\delta^{18}\text{O}_{\text{benthic}}$ and sea-level, which should be - and are - very, very correlated.

R: Thank you very much for your helpful comments. We have now cited relevant previous studies in Lines 358-361 and 407-410 to clarify the contributions of earlier work:

“Our results indicate that sea-surface temperature and eustatic sea-level changes correlate with fluctuations in richness for planktonic³³ and larger benthic foraminifera over the entire 28-Myr study interval²⁷.”

“Our results indicate that sudden PF and LBF richness decreases near the Eocene/Oligocene boundary may be related to the combination of sea-surface temperature decrease and eustatic sea-level fall^{27,28} (Fig. 3).”

While we agree that $\delta^{18}\text{O}_{\text{benthic}}$ has been widely used to reflect both temperature and sea-level changes (as in Ezard et al., 2011), our approach is to treat these environmental variables separately. Specifically, we incorporate independent records for eustatic sea level (Miller et al., 2020), deep-sea temperature (Cramer et al., 2009), and sea-surface temperature (Auderset et al., 2022) to disentangle their respective relationships with richness patterns of different foraminiferal groups. This allows us to examine the correlations individually—an aspect that adds a novel perspective.

In light of your comment on collinearity, we have removed the explicit correlation with $\delta^{18}\text{O}_{\text{benthic}}$, as it is indeed highly collinear with sea-level data in our dataset. The variance inflation factor (VIF) for $\delta^{18}\text{O}_{\text{benthic}}$ was 19.6, well above the commonly accepted threshold, whereas all other variables had VIFs below 10. We added the description in Lines 697-699:

“However, the $\delta^{18}\text{O}_{\text{benthic}}$ is not used for correlation, as its variance inflation factor (VIF) is larger than 10, indicating potential multicollinearity with other proxies, such as sea level and temperature.”

237-239: The phrasing here is awkward.

R: We have modified the expression as follows (please see Lines 411-412):

“SBF have different ecological features and therefore show a different extinction pattern across the Eocene/Oligocene boundary, compared with PF and LBF.”

304-305: I understand the authors intent here, but this statement is overly grandiose. One could say that studying the pollen found in annual varves in a lake is a much finer resolution than found here, as just one example.

R: Thank you for pointing out this. This is a finer resolution in deep time. We have revised the expression to make it more accurate. Please see Lines 484-486:

“This study is the first that links environmental change and biodiversity history at a fine temporal resolution over a long-time scale (~28 Myr) in deep time.”

Supplemental

L189: I think this "dig more deeply" ability of CONOP.EA is the recombination, but it is unclear as written.

R: Thank you for this comment. Yes, CONOP.EA can dig more deeply in the solution space because of the inclusion of the recombination algorithm. We have revised the wording in Lines 646-648 to improve clarity:

“...gives CONOP.EA a unique ability to dig further to find a more globally optimized

sequence by recombining various optimal sequences from previous computations.”

Reviewer #1 (Remarks on code availability):

I reviewed the dataset, but not the code.

Reviewer #2 (Remarks to the Author):

Lu et al present an interesting, well written manuscript exploring the relationship between foraminiferal richness and various abiotic parameters using a novel AI algorithm. This manuscript is unique in that it combines both planktonic and benthic foraminifera something rarely done in the community. Using the newly compiled dataset and AI algorithm the authors show that foraminifera show different responses to the Eocene-Oligocene climatic transition depending on their main ecology with small benthic foraminifera showing opposing trends to the other groups in the late Eocene which the authors hypothesize to be a result of increased export of organic matter to the deep.

R: We sincerely thank you for your positive and thoughtful comments. We truly appreciate your recognition of the novelty of the new AI algorithm and our efforts to analyze foraminiferal responses across groups with distinct ecological features.

This manuscript is well written and provides a novel approach to a longstanding problem, but in it's current form does not emphasize enough why this manuscript is novel and unique.

R: Thank you very much for this important comment. Another reviewer also raised a similar point. By following the advice of the reviewers and editor, we have restructured the manuscript to emphasize its novelty and uniqueness more clearly. Please see our detailed responses below.

The authors spend a lot of time explaining that the species curves generated show remarkable resemblance to those already produced, despite the AI algorithm producing a higher-resolution dataset, I would therefore encourage the authors to take some time and explain why this dataset and results are unique and what the differences between this and published datasets reveal. I believe reframing the manuscript and addressing the minor points below will improve the manuscript for publication in this journal.

R: We truly appreciate your valuable suggestion. In the earlier version of the manuscript (originally prepared for *Nature*), strict length limitations prevented us from elaborating on several important points. During this revision process, we reframed the manuscript and incorporated comprehensive explanations of the advantages of the new dataset and methods in the main text. First, we provided a detailed description of the characteristics of the new dataset. For example, it encompasses three groups of foraminifera with different ecological types, has a broader geographical distribution, and contains a greater number of species. The number of foraminifera in each group is greater than in previous, similar studies. We have also outlined the core features of the new AI algorithm, explaining why it is faster and more efficient than previous versions. This additional content can be found in the new section titled ‘Reconstructing high-resolution chronology using CONOP.EA: an AI-inspired metaheuristics algorithm’ (see Lines 133–249 in the revised manuscript). We have also addressed all of the minor points that you raised below. Thank you again for your constructive feedback, which has helped us significantly improve the clarity and focus of the manuscript.

We have also added descriptions of the previously published datasets, including their fossil group coverage, temporal resolution, and limitations, to provide a clear comparison with our new dataset. Please see Lines 102-111:

“While previous compilations have mostly focused on planktonic foraminifera³⁰⁻³², including analyses of their ecologically driven diversity dynamics³³ and comparisons with other microfossil groups such as diatoms and calcareous nannofossils^{32,34}, comparative studies involving benthic foraminifera—descendants of the same ancestral lineage but with distinct ecological characteristics—remain limited due to challenges such as age calibration difficulties, taxonomic inconsistencies, and high levels of endemism. Furthermore, most richness reconstructions rely on binning schemes with uneven or highly variable temporal resolutions^{30,35,36}, though a few studies have applied unbinned methods³³.”

Moderate changes:

1. Line 73 – the term ecogroups here is slightly confusing in the planktonic foraminifera community ecogroups generally correspond to stable isotope defined depth habitats (see Aze et al, 2011) whilst here the authors refer to much larger delineations. I suggest the authors reword this section to avoid confusion. In addition, I would recommend bringing some information regarding the 3 groups from the Supplementary Material into this section to explain habitat affinities of LBF and SBF and why they are separated as currently this separation seems arbitrary and based on size alone.

R: Thank you for your valuable suggestions. To avoid confusion with the established usage of “ecogroups” in the planktonic foraminifera community, we have replaced the term with “groups” to refer to the broader classifications used in our study. Additionally, we have added descriptions about the habitat affinities of PF, LBF, and SBF and how they are separated based on ecological and environmental characteristics rather than size alone. Please see Lines 76-93:

“Foraminifera are single-celled protists that can be separated into three groups that are generally distinct in their life-history strategy, morphology, and ecology, namely the planktonic foraminifera (PF), larger benthic foraminifera (LBF) and small benthic foraminifera (SBF). PF are free-floating and primarily inhabit open ocean waters, with their distribution strongly influenced by sea surface temperature, salinity, oxygen levels, nutrient availability, and light intensity. These factors not only affect their survival and vertical distribution but also their symbiotic relationships with photosynthetic organisms. SBF, on the other hand, are more ecologically versatile, occupying a broad range of depths from shallow to deep marine settings. Their survival is primarily governed by dissolved oxygen levels and food availability, and historical fluctuations in these parameters have been linked to major extinction events in foraminiferal history. In contrast, LBF are predominantly found in warm, shallow tropical to subtropical marine environments and are characterized by complex internal structures. Their ecological range is narrower than that of SBF and it is controlled by light intensity,

nutrient availability, and salinity because of their reliance on phototrophic symbionts (See SI). They are abundant, geographically widespread and sensitive to environmental changes^{18,19}.”

2. Line 86-68 – “Further, compilations have only involved one group of foraminifera” – expand on this point as to why and what group of foraminifera the authors are referring, I assume planktonic from the references, why are one group focused on more than others?

R: Sorry for the confusion. Yes, we are referring to planktonic foraminifera (PF). In general, data on benthic foraminifera (BF) are less readily available due to various challenges, including difficulties in age calibration, taxonomic inconsistencies, and high levels of endemism. To clarify this point, we have revised the wording in Lines 102-108 as follows:

“While previous compilations have mostly focused on planktonic foraminifera³⁰⁻³², including analyses of their ecologically driven diversity dynamics³³ and comparisons with other microfossil groups such as diatoms and calcareous nannofossils^{32,34}, comparative studies involving benthic foraminifera—descendants of the same ancestral lineage but with distinct ecological characteristics—remain limited due to challenges such as age calibration difficulties, taxonomic inconsistencies, and high levels of endemism.”

3. Figures 1- 2 – A lot of the text refers to very small intervals throughout, some within the individual stages, these are hard to pinpoint due to the lack of minor tick marks or other visual aid. Consider adding something to both plots. An effort to do this has been done in Figure 1, delineating event numbers but these are then not referred to in the text.

R: Thank you very much for this helpful suggestion. We have now added minor tick marks to both figures to improve readability and facilitate identification of the small intervals. We have also added citations to the relevant event numbers in the main text when discussing them. For example, in Lines 261-264:

“Foraminifera display an increasing species richness pattern in the early and middle Lutetian (Event No.1 in Fig. 2), followed by the Bartonian richness decline (BRD, Event NO.2 in Fig. 2), including a long-term decrease from the late Lutetian to earliest Priabonian (Fig. 2a–c).”

Minor Changes:

4. Line 57 – p in pCO₂ should be italicized

R: Thank you for your suggestion. We have corrected this.

5. Line 103 – 06 – this sentence is difficult to read, consider rewording

R: Thank you for pointing out this. We have rewritten this sentence. Please see Lines 255-258:

“From 48 Ma to 20 Ma, foraminiferal richness shows notable fluctuations, characterized by two phases of increase and two of decline (Fig. 2a–c). In addition, each foraminiferal group exhibits a distinct richness trajectory (Fig. 3a).”

6. Line 117 – Include a figure reference here

R: Thank you for your suggestion. We have included the citation of the relevant figure in Lines 273-275, as follows:

“The Eocene–Oligocene richness crisis is evident as a long-term richness reduction from the late Priabonian to the earliest Chattian (from 35.06 ± 0.41 Ma to 25.59 ± 0.66 Ma; Fig. 2a).”

7. Line 274 – Is LIP defined before this point? If not then define it here.

R: Thank you. The term LIP is first defined in Lines 68-69.

Methodology

I think the manuscript would benefit from more information regarding the dataset used such as:

Were the occurrences taken from publications or ODP/IODP reports and how were these chosen?

What is the resolution differences between the groups?

What is the geographical spread of the data?

R: Thank you for your suggestions.

Firstly, regarding the source of the data, all of the data come from peer-reviewed journal articles and scientific reports. We have added the following descriptions of the dataset in Lines 156-160:

“Occurrence data were extracted from original biostratigraphic distribution records (depth-versus-taxon) reported in peer-reviewed journal articles and scientific reports, including 12 continental drill cores, 48 DSDP/ODP drill cores, five ocean drill cores, and 96 outcrops.”

Secondly, regarding the resolution differences between groups, we calculated the average number of independent biostratigraphic events per section for each group—that is, fossil occurrences that do not coincide with any other taxon at the same stratigraphic level. A higher average indicates finer temporal resolution, because more unique events allow us to distinguish shorter time intervals, vice versa. Using this metric, we selected only the relatively dense sections—those containing more than 30 biostratigraphic events—for analysis, in order to reduce potential sampling biases. Based on the average number of independent events per section, planktic foraminifera (PF) exhibit the highest resolution (0.3037 level per section), followed by small benthic foraminifera (SBF) at 0.2441 level per section. Large benthic foraminifera (LBF) show the lowest resolution, with a value of 0.08 level per section. These differences were considered when interpreting temporal patterns and comparing evolutionary rates across groups.

We added the corresponding description in Lines 577-588 as follows:

“To ensure that our subsequent analyses are based on data with sufficient temporal resolution and reliability, we first evaluated the stratigraphic resolution of PF, SBF, and LBF. A pre-estimated result shows that PF exhibited the highest resolution (0.3037 levels per section), followed by SBF at 0.2441 levels per section, while LBF demonstrated the lowest resolution (0.08 levels per section). This resolution is estimated from the average number of independent bioevents per section—that is, fossil occurrences that do not coincide with any other taxon at the same stratigraphic level. A higher average indicates finer temporal resolution, because more unique events allow us to distinguish shorter time intervals, and vice versa. To minimize sampling bias, we included in our analysis only stratigraphic sections containing more than 30 biostratigraphic events, thereby ensuring sufficient data density.”

Third, regarding the geographical spread of the data, we have added a description of the geographic distribution of the data in Lines 153-156. We have also revised Fig. S1 to include colored circles indicating which foraminiferal groups (SBF, LBF or PF) are represented at each site or section, so that this figure can show simple geographic distribution of different foraminifera groups. Please see Lines 153-156:

“Of the 161 sections/sites, 102 yield one foraminiferal group, 59 yield at least two groups, and 16 include all three groups, covering major ocean basins, including the Atlantic and Indian Oceans, as well as key marginal seas such as the Mediterranean, Caribbean, and Gulf of Mexico (Fig. S1).”

Reviewer #3 (Remarks to the Author):

The manuscript describes a global trajectory of foraminiferal species during the Eocene–Oligocene transition. To do so, it relies on a rich foraminifera fossil record (~45 k fossil occurrences, ~9 k first and last occurrence, ~1.2 k species). The manuscript also proposes a new metaheuristics optimization algorithm (constrained optimization based on evolutionary algorithm, CONOP.EA), improving on CONOP.SAGA. The manuscript is generally well organized and well written, with appropriate figures and tables. I have only minor suggestions.

R: Thank you very much for taking the time to review our manuscript and for your positive evaluation. We have revised the manuscript according to your suggestions. As follows, we provide detailed responses to each of the points you raised.

1. My only main comment is that I suggest being a bit more specific in the abstract and conclusions that the proposed methodology is based on a metaheuristic optimization algorithm, or an optimization tool that uses AI-inspired metaheuristics. A “novel AI algorithm” is slightly vague, especially now that the term AI is most frequently associated with machine (or deep) learning.

R: Thank you for your suggestions. We have changed it to “AI-inspired metaheuristics algorithm” in the entire manuscript to avoid any potential misunderstandings.

The overall methodology seems sound, however I had trouble using the provided executables as described below.

R: We apologize for the confusion and appreciate your efforts in trying to run the program. We have now included more detailed instructions on how to execute the provided files in the ReadMe.txt file. We hope that the updated guidance will enable you to run the program smoothly. You are welcome to contact us if you have any further questions.

Minor comments:

Supporting information

2. Lines 199-200: “In the meanwhile, the average time costs of CONOP.SAGA and CONOP.EA are 40944.83s and 2387.71s, respectively” – I suggest converting these numbers to a more human readable format (hours, minutes, seconds). I also suggest providing information on what CPU was used. Similar for table S2.

R: Thank you for this suggestion. We have modified the expressions in Lines 620-626 as follows:

“In the meantime, the average wall-clock time costs of CONOP.SAGA and CONOP.EA are 13h:09m:30s and 33m:56.56s, respectively, on an Intel Core i9-10900F processor with 10 cores and a base clock speed of 2.80 GHz, equipped with 32 GB of DDR4 RAM. The performance result demonstrates that CONOP.EA ran ~23 times faster than CONOP.SAGA (estimated over 40,000 trials) by using a renewed function “newpen”,...”

Additionally, a note for table S2 is added in Supplementary Lines 650-651 as follows:

“Note: Computations are conducted on the ‘Tianhe II’ Supercomputer with 4 nodes (256 CPU cores), and 1 TB of user file-system space.”

3. Lines 286-287: “We first ran CONOP.SAGA on the ‘Tianhe II’ supercomputer” – I suggest adding more information about the resources used (number of cores used, CPU speed, memory required if easy to estimate, etc).

R: Thank you for your suggestion. We have added detailed descriptions in Supplementary Lines 340-346 as follows:

“The Tianhe II supercomputer comprises 16,000 compute nodes, each equipped with two Intel Xeon E5-2692 v2 12-core CPUs operating at a base clock of 2.2 GHz (turbo up to 3.0 GHz) and three Intel Xeon Phi 31S1P co-processors—yielding a total of 3,120,000 CPU cores and 48,000 coprocessor cards—backed by 1,375 TiB of DDR3 system memory and a 12.4 PB global storage system. Our project, requesting 4 nodes (256 CPU cores), was provisioned with 1 TB of user file-system space.”

4. File EOT_forams_Readable.csv (<https://doi.org/10.5061/dryad.jh9w0vtk5>)

It might be my computer, but question marks appear in some column names when I open the file using text editors (e.g., VS Code, or notepad++, two different computers, Sark?y vin 13_base,Sark?ySark?y vin 13_top,Ke?ili vin 15_base,Ke?ili vin 15_top) so I wonder if there was some encoding conflict when uploading the files or if that was intended?

R: We apologize for the incorrect headers caused by the encoding conflict. These have now been corrected and standard characters are being used to prevent any issues with interpretation in the new data file 'EOT_forams_Readable.csv'.

Executable CONOP++.exe

5. Unfortunately, I was not able to run the program. I use another SO for work and I only have access to an old Windows laptop, so I don't know if that's what's causing the issue. However, the program did not seem to take my input in the "Enter annealing" input after selecting "Project > Run - TXT". The program window goes black I select "Project > Run - CHT".

R: We apologize for the problem you encountered. Based on your description, it appears that you have successfully launched the CONOP program but got frozen with the large parameters set previously.

To enhance compatibility across different computing environments and further improve user experience, we have updated the program. Please use the new version in the software package "EOT_forams_CONOP_r1.zip". Detailed information about the program can be found in the "ReadMe.md" or "ReadMe.txt" files. Below, we provide a brief introduction:

The zip file contains two main folders: "CONOPEA_executable" and "CONOPEA_source code". The first folder contains the executable program and the second folder contains the source code. For your convenience of using, we have provided an executable program with two core modes for CONOPEA in the

“CONOP.EA_executable” folder:

1. Optimization from scratch (“CONOP_EA_from_scratch.exe”): This runs CONOP.EA from the initial CONOP input data file to find an optimal sequence.

2. Optimization from existing solutions (“CONOP_EA_from_existing.exe”): This starts the optimization process from pre-existing, high-quality solutions. For your convenience, we have pre-stored these solutions in the fold “benchmarks\CONOPLib_2574”, and you can run “CONOP_EA_from_existing.exe” directly.

We have tested the executables of these programs on multiple standard Windows PCs with different system versions, and all tests ran successfully.

To help you get started quickly, the default values of the three core parameters are intentionally set to relatively low levels: Initial temperature (Startemp) = 500, Steps = 700, and Trials = 60. This enables the program to rapidly load data and initialize calculations. You can modify these parameters by editing the “configs\config.yaml” file in folder (using a text editor) to match the values used in this study: Initial temperature (Startemp) = 500, Steps = 700, and Trials = 2,400,000. Note that using these settings will significantly increase the runtime. Please feel free to contact us if you have any further questions.

Reviewer #3 (Remarks on code availability):

6. I am not sure if the code itself is available, it seems only the executable is. There is a README file, but I could not use the executable as described in the main review.

R: Thank you for your comment. We have upgraded both the programme and the latest executable source code for the package. You are welcome to contact us if you have any further questions.

Thank you very much for carefully reviewing our manuscript and providing valuable comments. We hope that our revisions and responses address all of your concerns.

REVIEWER COMMENTS

Reviewer #1 (Remarks to the Author):

First, let me apologize for the exceedingly slow time for this second review. I got covid twice in 2 months, the first of which was the literal first day of the term. Thus, I've spent the last several months catching up with work.

R: We sincerely thank you for taking the time to conduct this second-round review under such difficult circumstances. We are very sorry to hear about your recent illness, and we hope that you are feeling much better now.

I was really happy to see Lu et al. 2025 published! Felt like a needed piece of this study's puzzle.

R: Thank you very much for your positive comment. We are glad to hear that you read our recent paper published in *Palaeogeography, Palaeoclimatology, Palaeoecology*, and that you found it significant as the technical foundation of this study.

Once the criticism discussed below relating to the Triton data is dealt with, I am fine with this work being published, but that issue must be addressed first.

R: Thank you for your positive assessment of the work and for raising this point. We appreciate the opportunity to clarify that the dataset analysed in this study consists solely of manually curated records digitised by the OneStratigraphy team, and does not include any entries derived from Triton or other databases. Further details are provided in the responses below, and the manuscript's wording has been refined to present the relevant information straightforwardly and clearly.

The authors have done a commendable job dealing with my criticisms, and while I have a few minor things that I would quibble with, some of those are outside the scope of this work. As an example, with respect to syn/diachrony: I understand the rebuttal, but I still fundamentally question the ability to claim having a temporal resolution so fine - my read on this as an (occasional, these days) biostratigrapher is that this claims we now know that FAD G. species A occurs at $x \text{ Ma} \pm 0.029 \text{ Myr}$ (0.0145 Myr?). I would suspect in some cases it's possible the dispersal of new planktic species could not ever be resolved as finely. But, again, I do not think this is a criticism that should hold this paper back at all, it's more of a fundamental 'can we actually state confidence levels that small like this given this is a biological system aliased by a slow geologic record' than anything to do with this paper fundamentally.

R: Thank you very much for your comment. We appreciate your broader perspective regarding the limits of temporal resolution in the stratigraphic record. In this study, the fine-scale series reflects the composite temporal resolution that is imputed by

algorithmically integrating large numbers of local biostratigraphic events through CONOP's optimisation framework. This enhanced resolving power is a feature of some quantitative biostratigraphic sequencing methods (Shaw, 1964; Sadler et al., 2009; Gradstein & Agterberg 2022) and has been well achieved in some recent studies, including Neogene diatoms (Cody et al., 2008), Paleozoic invertebrates (Fan et al., 2020), and Proterozoic eukaryotes (Tang et al., 2024). However, this concept differs from that of high-resolution geochronology dating methods such as CA-ID-TIMS.

To avoid any possible misunderstanding, we have rewritten relevant statements in the manuscript concerning this aspect, explicitly stating that this represents an imputed temporal resolution through the CONOP optimisation framework and quantitative biostratigraphic sequencing methods by using all fossil occurrence data. It is fundamentally distinct from the concept of absolute age-based temporal resolution derived from isotopic dating techniques.

The detailed revisions can be seen below.

Lines 136-139: *“This method aims to estimate the most complete range of species among sections/cores in the study dataset, thereby providing an imputed, algorithm-based temporal framework for high-resolution analysis of richness patterns in life-evolution studies^{37,43,44}.”*

Lines 246-250: *“It should be noted that this represents an imputed temporal resolution, which is conceptually distinct from the absolute age resolution obtained through isotope dating techniques such as chemical abrasion - isotope dilution - thermal ionisation mass spectrometry (CA-ID-TIMS).”*

Thank you for the guidelines on how to retrieve the deeper occurrence information, this is substantially different than what I was looking at before. Having poked through a few of the different entries, yes, the original publication is there - not sure how I must have ended up wherever I was initially in the database. Thanks!

R: Thank you for carefully examining the occurrence records in the database. We are glad that our explanation was helpful and that you were able to locate the original publications within the database.

The key criticisms remaining which must be addressed, however, is: What is the number of entries which came originally from the Triton or NSB import? The Triton import would also bring along the NSB and ForCens data with it, as an incomplete set of those data are contained within Triton. When I use the Triton reference in OneStratigraphy to find all the included Triton data, it pulls up 31,539 pages of occurrences, though it's unclear if those were included in this work, as the first page appears to be missing key information for it to be usable (locality is unknown). With a full "foraminifera" search providing 56,495 pages, a rough estimate of the Triton component of this dataset could be on the order of ~55%.

R: Thank you for raising this important point. To clarify at the outset, no entries derived from Triton, NSB, or ForCenS were used in this study; all analysed data come from records manually digitised by the OneStratigraphy team from the literature. Further context is provided below.

Using the OneStratigraphy interface (accessed 27 Nov 2025), a search using the Triton reference returns 461,614 foraminiferal occurrences (~30,775 pages), with the reference field labelled “Fenton et al., 2021” (which, following your helpful suggestion, has been corrected to “Fenton & Woodhouse et al., 2021”). For comparison, the manually curated and validated foraminiferal records in OneStratigraphy currently contain 394,130 occurrences (~26,276 pages), each with the original literature source listed in the “Reference” field. Page numbers may vary slightly depending on display settings. These numbers indicate that Triton-derived entries presently account for ~53.9% of the whole foraminiferal records in the database—very close to your estimate of ~55%. Notably, the manually curated foraminiferal records in OneStratigraphy are concentrated within specific time intervals and research projects—such as the PETM and EOT—rather than providing comprehensive coverage of the whole Cenozoic.

You are right that the Triton data in OneStratigraphy lack locality information which is important for palaeobiologic studies. These missing fields stem from differences in metadata standards between Triton and OneStratigraphy, and we are currently working to reconcile them. For this reason, none of these records were included in the present study.

All of the occurrence data analysed in this study were manually curated and entered into OneStratigraphy by our team in 2020 and 2021, before the Triton data were imported. All of these manually curated records contain traceable provenance in the database. For example, the first entered section for this study (see Table S6), ‘Zongpubei’, has the Section ID SE202005060304540, while the last entered section, ‘Alano’, has the Section ID SE202110280146975. Using the OneStratigraphy interface (“Stratigraphic Data” → “Advanced Search” → “Section”) and querying these Section IDs in the “Section no.” field retrieves the corresponding section records. It can be found that the ‘Zongpubei’ section was digitised and entered into OneStratigraphy by Qin Chen on 6 May 2020, while the ‘Alano’ section was digitised and entered by Xiaohong Zhou on 28 Oct. 2021. In both cases, the original literature sources from which the stratigraphic information was digitised are documented in the “View reference” tab associated with each section record. ‘Zongpubei’ is digitised from the peer-reviewed publication entitled “Late Cretaceous to early Paleogene foraminiferal biozones in the Tibetan Himalayas, and a pan-Tethyan foraminiferal correlation scheme” by BouDagher-Fadel et al. (2015), while ‘Alano’ is digitised from the peer-reviewed publication entitled “Proposal for the global boundary stratotype section and point (GSSP) for the Priabonian stage (Eocene) at the Alano section (Italy)” by Agnini et al. (2021). The related information of bibliography, enterer and date of entry can be found

in Table S6.

Following your comments, we have added necessary descriptions in the main text and Supplementary Information. The detailed revisions are listed below.

Lines 143-149 in the main text: *“These records were manually digitised from the original published literature by the OneStratigraphy team. After the cleaning process (see SI), the final dataset comprises approximately 40,000 fossil occurrences of 1,269 species derived from 161 drill cores and outcrops (Fig. S1; Table S6).”*

Lines 531-538 in Methods of the main text: *“The dataset analysed here was manually digitised by the OneStratigraphy team between May 2020 and October 2021. It covers a wide range of published literature and therefore incorporates records documented under diverse reporting styles, formats, and taxonomic conventions. The corresponding reference sources are traceable in the OneStratigraphy database through the bibliographic information provided in Table S6. Prior to analysis, the dataset was first cleansed and standardised by excluding non-foraminifera fossils, addressing open nomenclature, correcting typographical errors, etc. (see SI).”*

Lines 75-89 in the Supplementary Information: *“Each stratigraphic section is assigned a unique Section ID in OneStratigraphy (Table S6), allowing direct traceability of data entry history, contributor identity, and original literature sources. For example, the first and last entered entries in Table S6 correspond to the sections ‘Zongpubei’ (Section ID: SE202005060304540) and ‘Alano’ (Section ID: SE202110280146975), respectively. Querying these Section IDs in OneStratigraphy (“Stratigraphic Data” → “Advanced Search” → “Section” → “Section no.”) retrieves the full section records. The ‘Zongpubei’ section was digitised and entered into OneStratigraphy by Qin Chen on 6 May 2020, while the ‘Alano’ section was digitised and entered by Xiaohong Zhou on 28 Oct. 2021. In both cases, the original literature sources from which the stratigraphic information was digitised are documented in the “View reference” tab associated with each section record. The related information of bibliography, data enterer and date of entry can be found in Table S6. This structure ensures transparent and clear linkage of all input sections to their original published sources within the OneStratigraphy database.”*

It's possible that I'm looking at this incorrectly and there's certainly a potential for some miscommunication on my side here. Is all the Triton data within OneStratigraphy simply "staged" at the moment and not included in this analysis? Is some of it staged and some included, and if so, how much? The text in the rebuttal re: Triton "However these data will not be used in any formal studies until the metadata are fully completed and verified. The OneStratigraphy team is currently trying to enrich and correct the metadata, including linking each Triton record back to its original publication." From the original Triton publication: "The source of the data (**source**) is recorded based on the data citation and **year** in which it was collected. The primary data sources

(e.g. Neptune, Pangaea) are given unique IDs (**db.source**). Individual datasets within this are given unique IDs (**db.ID**); these are particularly relevant for Pangaea where multiple, separate datasets exist. Each site is given a unique **holeID**, and samples within sites are designated using the **sampleID** (which is a unique number added to the **holeID**). The **rowID** is created by combining the database ID, the **sampleID**, and a unique number assigned to each row (i.e. species). The **person** who entered the data and **date** of the most recent update of that entry is also recorded."

Required metadata (more, in fact) appears to be within the original Triton data, unless the authors are looking for something not included here? This leads me to believe that Triton is included, at least partially.

R: Thank you for raising this important point. All the Triton-derived records hosted in OneStratigraphy are in a “staged” (imported-but-unvalidated) state, and none of them were used in this study. The EOT dataset analysed here consists solely of manually digitised records from the literature by the OneStratigraphy team.

We apologise if our earlier wording caused any misunderstanding regarding the status of Triton data. We have great respect for the Triton project (and also prior projects involved such as NSB and ForCenS) as a comprehensive, well-documented, species-level database of Cenozoic planktonic foraminiferal occurrences. Its metadata structure, however, differs from that of OneStratigraphy, which is designed as a stratigraphic framework and therefore requires more fields, such as lithologic units, documented hiatuses or disturbances, and detailed locality specifications. Some of these fields fall outside the scope of Triton, and others require additional harmonisation during database integration. As a result, the Triton-derived records currently present in the OneStratigraphy database need further metadata reconciliation on our side—not a reflection of any issue with Triton itself. Accordingly, none of Triton-derived records were included in the analyses presented in this study. Once this process is complete, we sincerely hope that OneStratigraphy can facilitate broader community access to and citation of Triton, and serve geologists worldwide alongside the independently curated records in OneStratigraphy.

The 55% is obviously a rough and again, I have no doubt, incorrect number, but even some data incorporated here from NSB or Triton is present, the work being built upon must be discussed, acknowledged, and further outlined how the OneStratigraphy data is unique within the text. Especially given this study's comparison to the Fenton and Woodhouse richness curve. If the majority of the underlying data were from Triton, it's no surprise the richness curves were similar. As far as I can tell, there is no discussion of Triton or Neptune by name in the paper, which then relegates them to simply citations for criticisms of them as previous work (e.g., L99-106) and comparison points, rather than as major sources of data, or the major source of data if the 55% is correct. I'd also like to point out that the Fenton et al. paper is a joint first author paper, and so should be cited as Fenton & Woodhouse et al. Further, one of the major sources of information

in the Triton dataset is ForCenS, which does not appear in the publication at all.

R: Thank you for this thoughtful comment. All the data analysed here were manually digitised by the OneStratigraphy team from original literature with traceable provenance information. Triton, NSB, and ForCenS do not constitute data sources for the present study. Therefore, the results presented here represent an independent analysis. The similarity of our curves to those based on previous databases further supports the reliability of our findings.

We apologise for any unintended ambiguity. The passage in lines 99–106 was not intended as a criticism of Triton, Neptune, or any specific compilation; rather, our intention was to summarise the long-standing methodological challenges associated with large-scale fossil synthesis studies that predate and extend beyond any individual database.

Following your comments, we have revised the description in the main text. The detailed revisions can be seen below.

Lines 531-536 in Methods of the main text: *“The dataset analysed here was manually digitised by the OneStratigraphy team between May 2020 and October 2021. It covers a wide range of published literature and therefore incorporates records documented under diverse reporting styles, formats, and taxonomic conventions. The corresponding reference sources are traceable in the OneStratigraphy database through the bibliographic information provided in Table S6.”*

Lines 98-109: *“While previous large-scale studies and compilations have mostly focused on planktonic foraminifera³⁰⁻³², including analyses of their ecologically driven diversity dynamics³³ and comparisons with other microfossil groups such as diatoms and calcareous nannofossils^{32,34}, comparative studies involving benthic foraminifera—descendants of the same ancestral lineage but with distinct ecological characteristics—remain limited, largely reflecting field-wide challenges inherent to large-scale synthesis (e.g., age calibration difficulties, taxonomic inconsistencies, and high levels of endemism). Furthermore, most richness reconstructions rely on binning schemes with uneven or highly variable temporal resolutions^{30,35,36}, though a few studies have applied unbinned methods³³.”*

We thank you for noting the citation format. We have corrected the citation to “Fenton & Woodhouse et al.” accordingly in the main text (Line 302), References (Line 745), and the legend in Fig. S7b.

R1.2 of FAIR is “Metadata are associated with detailed provenance.” I understand the difficulty of producing a way of presenting data which has a “NSB->Triton->OneStrat” or “ForCenS->Triton->OneStrat” provenance, or which literature source NSB/Triton pulled data from. A discussion of where this data came from as well as the amount of

the data from those sources is important, and any modification of those individual sources.

R: Thank you for raising this important point regarding the FAIR principles and data provenance. We fully agree that “where this data came from as well as the amount of the data from those sources is important”. Accordingly, these Triton-derived data imported into OneStratigraphy are being further reconciled, as you recommended. However, since we did not use any entries from Triton, NSB or ForCenS, all the data analysed in this study were digitised manually from the literature by the OneStratigraphy team. Therefore, we have clarified this point in the manuscript by detailing the dataset size, data provenance, and data acquisition methods. We also provide the original bibliographic information and other traceable details in Table S6 to facilitate searching within the OneStratigraphy database. The detailed revisions can be seen in Lines 140-149:

“Our assembled global foraminifera dataset (raw dataset) comprises 13,138 local bioevents records (i.e., first and last appearance records) and ~60,000 occurrences of 2,988 taxonomic units from 163 published stratigraphic sections, encompassing both calcareous and agglutinated foraminifera. These records were manually digitised from the original published literature by the OneStratigraphy team. After the cleaning process (see SI), the final dataset comprises approximately 40,000 fossil occurrences of 1,269 species derived from 161 drill cores and outcrops. (Fig. S1; Table S6).”

Lines 531-538 in Methods of the main text: *“The dataset analysed here was manually digitised by the OneStratigraphy team between May 2020 and October 2021. It covers a wide range of published literature and therefore incorporates records documented under diverse reporting styles, formats, and taxonomic conventions. The corresponding reference sources are traceable in the OneStratigraphy database through the bibliographic information provided in Table S6. Prior to analysis, the dataset was first cleansed and standardised by excluding non-foraminifera fossils, addressing open nomenclature, correcting typographical errors, etc. (see SI).”*

We have also added a brief acknowledgement in the revised manuscript to thank the research teams who originally studied and published the stratigraphic sections and fossils used in this study (Lines 851-854):

“We thank the numerous research groups whose published work underpins the stratigraphic sections and fossil occurrences used in this study. The full list of source references and section identifiers is provided in the Supplementary Information and Table S6.”

Without a full accounting of these concerns, it is not possible to recommend this manuscript for publication. If there is no Triton, NSB, or ForCenS data, or any other data compilation incorporated than I recommend this manuscript for publication.

R: Thank you for your comment. We fully understand your concern regarding the data

source and implementation of the FAIR principles. Since we did not use any data from Triton, NSB, ForCenS, or any other databases in this study, all data were independently collected from the literature by the OneStratigraphy team.

We hope that this clarification resolves your concerns and meets the conditions you outlined for recommending publication.

Thank you again for your comments and for your kind consideration of this manuscript. Your suggestions and comments during the two rounds of peer review have helped us to significantly improve the quality of the manuscript!

Reviewer #3 (Remarks to the Author):

I would like to thank the authors for addressing my comments, I have no further suggestions.

R: We sincerely thank you for your careful evaluation and positive assessment of our revisions. We appreciate your supportive feedback.

Reviewer #3 (Remarks on code availability):

The executable CONOP_EA_from_existing.exe worked on my computer (Windows 10, Intel Core i5-7300HQ CPU @ 2.5Ghz, 8 Gb RAM)

R: Thank you for confirming that the executable runs successfully on your system.